# Stabilizing Contrastive RL: Techniques for Robotic Goal Reaching from Offline Data

**Chongyi Zheng**[1]     **Benjamin Eysenbach**[2]     **Homer Walke**[3]     **Patrick Yin**[4]

**Kuan Fang**[5]     **Sergey Levine**[3]     **Ruslan Salakhutdinov**[1]

[1]Carnegie Mellon University     [2]Princeton University     [3]UC Berkeley
[4]University of Washington     [5]Cornell University

`chongyiz@andrew.cmu.edu`

## Abstract

Robotic systems that rely primarily on self-supervised learning have the potential to decrease the amount of human annotation and engineering effort required to learn control strategies. In the same way that prior robotic systems have leveraged self-supervised techniques from computer vision (CV) and natural language processing (NLP), our work builds on prior work showing that the reinforcement learning (RL) itself can be cast as a self-supervised problem: learning to reach any goal without human-specified rewards or labels. Despite the seeming appeal, little (if any) prior work has demonstrated how self-supervised RL methods can be practically deployed on robotic systems. By first studying a challenging simulated version of this task, we discover design decisions about architectures and hyperparameters that increase the success rate by $2\times$. These findings lay the groundwork for our main result: we demonstrate that a self-supervised RL algorithm based on contrastive learning can solve real-world, image-based robotic manipulation tasks, with tasks being specified by a single goal image provided after training.[1]

## 1 Introduction

Self-supervised learning serves as the bedrock for many NLP and computer vision applications, leveraging unlabeled data to acquire good representations for downstream tasks. How might we enable similar capabilities for robot learning algorithms? In NLP and computer vision, self-supervised learning is typically done via one objective (often denoising), while the downstream tasks use a different objective (e.g., linear regression). In the RL setting, prior work has shown how a (self-supervised) contrastive learning objective can simultaneously be used to learn *(1)* compact representations, *(2)* a goal-conditioned policy, and *(3)* a corresponding value function (Eysenbach et al., 2022). From a robotics perspective, this framing is appealing because users need not manually specify these components (e.g., no tuning hyperparameters in a reward function).

Applying this self-supervised version of goal-conditioned RL to real-world robotic tasks faces several challenges. First, because much of the work on self-supervised learning has been focused on CV and NLP, the recent innovations in those field may not transfer to the RL setting. Second, the right representations for the RL setting will need to capture nuanced and detailed information not only about the environment dynamics, but also about what states may lead to certain goal states. This is in contrast to typical representations from computer vision, which mostly reason about the relationship between objects in a static image (e.g., for classification or segmentation). Nonetheless, this challenge presents an opportunity: the representations learned via a self-supervised RL algorithm might capture information not just about images themselves, but about the underlying *decision making problem*; they may tell us not only what is contained within an image, but how to generate a plan for reaching other images.

The aims of this paper are to *(1)* build an effective goal-conditioned RL method that works on the real-world robotics datasets, *(2)* understand the differences between this method and prior approaches, *(3)* and empirically analyze why our method works better. While the datasets we will use are big from a robotics perspective, they are admittedly much smaller than the datasets in computer vision

---

[1]Project website: `https://chongyi-zheng.github.io/stable_contrastive_rl`.

and NLP. We will focus on a class of prior RL methods based on *contrastive RL* (Touati & Ollivier, 2021; Eysenbach et al., 2022; 2020; Guo et al., 2018). With an eye towards enabling real-world deployment, we start by studying how architecture, initialization and data augmentation can stabilize these contrastive RL methods, focusing both on their *learned representations* and their *learned policies*. Through careful experiments, we find a set of design decisions regarding model capacity and regularization that boost performance by $+45\%$ over prior implementations of contrastive RL, and by $2\times$ relative to alternative goal-conditioned RL methods. We call our implementation **stable contrastive RL**. The key contribution is our real-world experiments, where we demonstrate that these design decisions enable image-based robotic manipulation. Additional experiments reveal an intriguing property of the learned representations: linear interpolation seems to corresponds to planning (see Fig. 6).

## 2  RELATED WORK

This paper will build upon prior work on self-supervised RL, a problem that has been widely studied in the contexts of goal-conditioned RL, representation learning, and model learning. The aim of this paper is to study what design elements are important for unlocking the capabilities of one promising class of self-supervised RL algorithms.

**Goal-conditioned RL.** Our work builds upon a large body of prior work on goal-conditioned RL (Kaelbling, 1993; Schaul et al., 2015), a problem setting that is appealing because it can be formulated in an entirely self-supervised manner (without human rewards) and allows users to specify tasks by simply providing a goal image. Prior work in this area can roughly be categorized into conditional imitation learning methods (Lynch et al., 2020; Ding et al., 2019; Ghosh et al., 2020; Srivastava et al., 2019; Gupta et al., 2020) and actor-critic methods (Eysenbach et al., 2020; 2022). While in theory these self-supervised methods should be sufficient to solve arbitrary goal-reaching tasks, prior work has found that adding additional representation losses (Nair et al., 2018; Nasiriany et al., 2019), planning modules (Fang et al., 2022a; Savinov et al., 2018; Chane-Sane et al., 2021), and curated training examples (Chebotar et al., 2021; Tian et al., 2020) can boost performance. Our paper investigates where these components are strictly necessary, or whether their functions can emerge from a single self-supervised objective.

**Representation Learning in RL.** Learning compact visual representations is a common way of incorporating offline data into RL algorithms (Baker et al., 2022; Ma et al., 2022b; Nair et al., 2022; Sermanet et al., 2018; Yang & Nachum, 2021; Such et al., 2019; Wang et al., 2022; Annasamy & Sycara, 2019). Some prior methods directly use off-the-shelf visual representation learning components such as perception-specific loss (Finn et al., 2016; Lange & Riedmiller, 2010; Watter et al., 2015; Stooke et al., 2021; Shridhar et al., 2023; Goyal et al., 2023; Gervet et al., 2023) and data augmentation (Laskin et al., 2020b;a; Yarats et al., 2021; Raileanu et al., 2021) to learn these representations; other work uses representation learning objectives tailored to the RL problem (Ajay et al., 2020; Singh et al., 2020; Zhang et al., 2020; Gelada et al., 2019; Han et al., 2021; Rakelly et al., 2021; Florensa et al., 2019; Choi et al., 2021). While these methods have achieved good results and their modular design makes experimentation easier (no need to retrain the representations for each experiment), they decouple the representation learning problem from the RL problem. Our experiments will demonstrate that the decoupling can lead to representations that are poorly adapted for solving the RL problem. With proper design decisions, self-supervised RL methods can acquire good representations on their own.

**Model Learning in RL.** Another line of work uses offline datasets to learn an explicit model. While these methods enables one-step forward prediction (Ha & Schmidhuber, 2018; Matsushima et al., 2020; Wang & Ba, 2019) and auto-regressive imaginary rollouts (Finn & Levine, 2017; Ebert et al., 2018; Nair et al., 2018; Yen-Chen et al., 2020; Suh & Tedrake, 2021) that is a subset of the more general video prediction problem (Mathieu et al., 2015; Denton & Fergus, 2018; Kumar et al., 2019), model outputs usually degrade rapidly into the future (Janner et al., 2019; Yu et al., 2020b). In the end, fitting the model is a self-supervised problem, actually using that model requires supervision for planning or RL. Appendix A compares these prior methods.

## 3 CONTRASTIVE RL AND DESIGN DECISIONS

In this section we introduce notations and the goal-conditioned RL objective, and then revise a prior algorithm that we will use in our experiments, mentioning important design decisions.

### 3.1 PRELIMINARIES

We assume a goal-conditioned controlled Markov process (an MDP without a reward function) defined by states $s_t \in \mathcal{S}$, goals $g \in \mathcal{S}$, actions $a_t$, initial state distribution $p_0(s_0)$ and dynamics $p(s_{t+1} \mid s_t, a_t)$. We will learn a goal-conditioned policy $\pi(a \mid s, g)$, where the state $s$ and the goal $g$ are both RGB images. Given a policy $\pi(a \mid s, g)$, we define $\mathbb{P}^{\pi(\cdot \mid \cdot, g)}(s_t = s \mid s_0, a_0)$ as the probability density of reaching state $s$ after exactly $t$ steps, starting at state $s_0$ and action $a_0$. The discounted state occupancy measure is a geometrically-weighted average of these densities:

$$p^{\pi(\cdot \mid \cdot, g)}(s_{t+} = s \mid s_0, a_0) \triangleq (1 - \gamma) \sum_{t=0}^{\infty} \gamma^t \mathbb{P}^{\pi(\cdot \mid \cdot, g)}(s_t = s \mid s_0, a_0). \tag{1}$$

The policy is learned by maximizing the likelihood of reaching the desired goal state under this discounted state occupancy measure (Eysenbach et al., 2020; Touati & Ollivier, 2021; Rudner et al., 2021; Eysenbach et al., 2022):

$$\mathbb{E}_{p_g(g)}[p^{\pi(\cdot \mid \cdot, g)}(s_{t+} = g)] = \mathbb{E}_{p_g(g)p_0(s_0)\pi(a_0 \mid s_0, g)} \left[ p^{\pi(\cdot \mid \cdot, g)}(s_{t+} \mid s_0, a_0) \right].$$

Following prior work (Eysenbach et al., 2022), we estimate this objective via contrastive representation learning (Gutmann & Hyvärinen, 2012); we will learn a critic function that takes a state-action pair $(s, a)$ and a future state $s_{t+}$ as input, and outputs a real number $f(s, a, s_{t+})$ estimating the (ratio of) likelihood of reaching the future state, given the current state and action. We will parameterize the critic function as an inner product between the state-action representation $\phi(s, a)$ and the future state representation $\psi(s_{t+})$, $f(s, a, s_{t+}) = \phi(s, a)^\top \psi(s_{t+})$, interpreting the critic value as the similarity between those representations.

Contrastive RL distinguishes a future state sampled from the average discounted state occupancy measure, $s_f^+ \sim p^{\pi(\cdot \mid \cdot)}(s_{t+} \mid s, a) = \int p^{\pi(\cdot \mid \cdot, g)}(s_{t+} \mid s, a) p^\pi(g \mid s, a) dg$, from a future state sampled from a arbitrary state-action pair, $s_f^- \sim p(s_{t+}) = \int p^{\pi(\cdot \mid \cdot)}(s_{t+} \mid s, a) p(s, a) ds da$, using the NCE-Binary (Gutmann & Hyvärinen, 2012; Ma & Collins, 2018; Hjelm et al., 2019) objective:

$$\mathbb{E}_{s_f^+ \sim p^{\pi(\cdot \mid \cdot)}(s_{t+} \mid s, a)} \underbrace{[\log \sigma(\phi(s, a)^\top \psi(s_f^+))]}_{\mathcal{L}_1(\phi(s, a), \psi(s_f^+))} + \mathbb{E}_{s_f^- \sim p(s_{t+})} \underbrace{[\log(1 - \sigma(\phi(s, a)^\top \psi(s_f^-)))]}_{\mathcal{L}_2(\phi(s, a), \psi(s_f^-))}. \tag{2}$$

This objective can also be formulated in an off-policy manner (see Eysenbach et al. (2020) and Appendix B); we will use this off-policy version in our experiments. Appendix C provides some intuition about a connection between contrastive RL and hard negative mining.

### 3.2 DESIGN DECISIONS FOR STABILIZING CONTRASTIVE RL

In this section, we describe the most important design factors to stabilize contrastive RL: *(i)* using appropriate encoder architecture and batch size, *(ii)* stabilizing and speeding up training with layer normalization and cold initialization, and *(iii)* combating overfitting with data augmentation.

**D1. Neural network architecture for image inputs.** In vision and language domains, scaling the architecture size has enabled large neural networks to achieve ever higher performance on ever larger datasets (Tan & Le, 2019; Brown et al., 2020; Dosovitskiy et al., 2020; Radford et al., 2021; Baker et al., 2022). While large vision models (e.g., ResNets (He et al., 2016) and Transformers (Vaswani et al., 2017)) have been adopted to the RL setting (Espeholt et al., 2018; Shah & Kumar, 2021; Kumar et al., 2022; Chen et al., 2021a; Janner et al., 2021), shallow CNNs remain pervasive in RL (Schrittwieser et al., 2020; Laskin et al., 2020a;b; Badia et al., 2020; Hafner et al., 2020; 2019), suggesting that simple architectures might be sufficient. In our experiments, we will study two aspects of the architecture: the visual feature extractor, and the contrastive representations learned on top of these visual features. For the visual feature extractor, we will compare a simple *CNN* versus a much larger *ResNet*. For the contrastive representations, we will study the effect of scaling the width

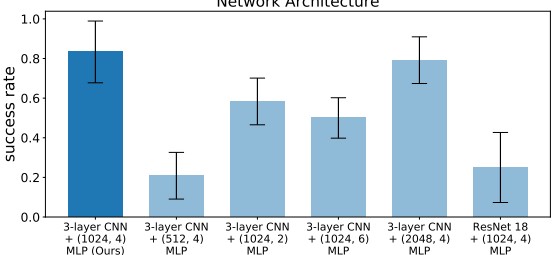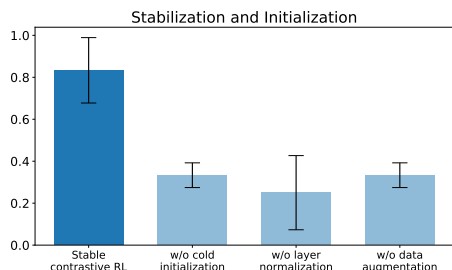

Figure 1: **Importance of architecture initialization, and normalization.** *(Left)* Using a deep CNN *decreases* performance but using a wide MLP is important. *(Right)* The good performance of our implementation depends on "cold initialization," layer normalization, and data augmentation. Prior work (Eysenbach et al., 2022) uses a small network architecture (256, 2) without the stabilization/initialization decisions.

and depth of these MLPs. While prior methods often train the visual feature extractor separately from the subsequent MLP (Radford et al., 2021), our experiments will also study end-to-end training approaches. Appendix D includes an overview of our network architecture.

**D2. Batch size.** Prior works in computer vision (Chen et al., 2020; He et al., 2020; Grill et al., 2020; Chen et al., 2021b) have found that contrastive representation learning benefits from large batch sizes, which can stabilize and accelerate learning. In the context of contrastive RL, using larger batches increases not only the number of positive examples (linear in batch size) but also the number of negative examples (quadratic in batch size) (Eysenbach et al., 2022). This means that algorithm would be able to see more random goals $s_f^-$ given a future goal $s_f^+$. We will study how these growing positive and negative examples as a result of growing batch sizes will affect contrastive RL.

**D3. Layer normalization.** We conjecture that learning from a *diverse* offline dataset, containing examples of various manipulation behaviors, may result in features and gradients that are different for different subsets of the dataset (Yu et al., 2020a). We will study whether adding layer normalization (Ba et al., 2016) to the visual feature extractor and the subsequent MLP can boost performance, following prior empirical studies on RL (Bjorck et al., 2021; Kumar et al., 2022). We will experiment with applying layer normalization to every layer of both CNN and MLP, before the non-linear activation function.

**D4. Cold initialization.** Prior work has proposed that the alignment between positive examples is crucial to contrastive representation learning (Wang & Isola, 2020). To encourage the alignment of representations during the initial training phase, we will weight initialization of the final feedforward layer. Precisely, we will initialize the weights in the final layers of $\phi(s, a)$ and $\psi(g)$ using $\mathrm{UNIF}[-10^{-12}, 10^{-12}]$, resulting in representations that remain close to one another during the initial stages of learning. We will compare this "cold initialization" approach to a more standard initialization strategy, $\mathrm{UNIF}[-10^{-4}, 10^{-4}]$.

**D5. Data augmentation.** Following prior work (Eysenbach et al., 2022; Fujimoto & Gu, 2021; Baker et al., 2022; Peng et al., 2019; Siegel et al., 2019; Nair et al., 2020; Wang et al., 2020), we will augment the actor objective to include an additional behavioral cloning regularizer, which penalizes the actor for sampling out-of-distribution actions. While most of our design decisions increase the model capacity, adding this behavioral cloning acts as a sort of regularization often important in the offline RL setting to avoid overfitting (Fujimoto & Gu, 2021). Initial experiments found that this regularizer itself was prone to overfitting, motivating us to investigate data augmentation (random cropping), similar to prior work in offline RL (Yarats et al., 2020; Laskin et al., 2020b; Hansen et al., 2021).

## 4 EXPERIMENTS

Our experiments start with studying design decisions that drive stable contrastive RL and use simulated and real-world benchmarks to compare contrastive RL to other offline goal-conditioned policy learning methods, including those that use conditional imitation and employ representation pre-trained by auxiliary objectives. We then analyze unique properties of the representations learned by stable contrastive RL, providing an empirical explanation for the good performances of our method. Finally, we conduct various ablation studies to test the generalizing and scalability of the policy learned by our algorithm. We aim our experiments at answering the following questions:

1. Which combination of those design factors is the most efficient one to drive contrastive RL in solving robotic tasks?

2. How does stable contrastive RL compare with prior offline goal-conditioned RL methods on simulated and real-world benchmarks?

3. Do representations learned by stable contrastive RL emerge properties that explain how it works?

4. Does the policy learned by our method benefit from the robustness and scalability of contrastive learning?

## 4.1 EXPERIMENTAL SETUP

Before discussing the experiments, we first introduce the experiment setups that we will use. Our experiments use a suite of simulated and real-world goal-conditioned control tasks based on prior work (Fang et al., 2022a;b; Ebert et al., 2021; Mendonca et al., 2021). *First*, we will use a benchmark of five simulated manipulation tasks proposed in (Fang et al., 2022a;b) (Fig. 2), which entail controlling a simulated robot arm to rearrange various objects. *Second*, we will use the goal-conditioned locomotion benchmark proposed in (Mendonca et al., 2021). These tasks entail reaching desired poses, such as controlling a quadruped to balance on two feet. *Third*, we will study whether good performance on these simulated benchmarks translates to real-world success using a 6-DOF WidowX 250 robot arm. We train on an expanded version of the Bridge dataset (Ebert et al., 2021) which entails controlling that robot arm to complete different housekeeping tasks. See Appendix E for details of the task and dataset and challenges in solving these tasks.

## 4.2 ABLATION STUDY

Our first experiment studies how different design decisions affect the performance of contrastive RL, aiming to find a set of key design factors (e.g., architectures, initialization) that boosts this goal-conditioned RL algorithm efficiently. We will conduct ablation studies on `drawer` (Fig. 1) and `push block, open drawer` (Appendix F.4) and report mean and standard deviations over three random seeds. The appendix contains additional experiments underscoring the importance of large batches (**D2.** Appendix F.13) and showing that lower-dimensional representations achieve higher success rates (Appendix F.15).

**D1. Network architecture.** We study the architectures for the image encoder and the subsequent MLP. For the image encoder, we compare a shallow *3-layer CNN* (similar to DQN (Mnih et al., 2015)) to a *ResNet 18* (He et al., 2016). Fig 1 *(Left)* shows that 3-layer CNN encoder outperforms ResNet 18 encoder by $2.89\times$ ($81\%$ vs. $28\%$), when incorporting same design decisions other than architectures. Additional experiments in Appendix F.1 suggest that this counterintuitive finding can likely be explained by overfitting. We next study the architecture of the subsequent MLP, denoted as (width, depth). Our experiments show that a $(1024, 4)$ MLP yields the best result, suggesting that wider MLPs perform better. In subsequent experiments, we use the 3-layer CNN as the visual feature extractor followed by a $(1024, 4)$ MLP. Appendix F.2 contains additional experiments on a "pretrain and finetune" setting, following prior work (Kumar et al., 2022; Nair et al., 2022) that uses visual encoder pretrained on vision datasets for RL tasks directly.

**D3. Stabilization, D4. initialization and D5. data augmentation.** We hypothesize that layer normalization balances feature magnitude and prevents gradient interference, while data augmentation mitigates overfitting. Our experiments in Fig. 1 *(Right)* suggest that both layer normalization and data augmentation (via random cropping) are critical for good performance.

We next study weight initialization, using the "cold initialization" strategy outlined in Sec. 3.2. We find that this strategy, which uses very small initial weights $\text{UNIF}[-10^{-12}, 10^{-12}]$, boosts performance by 2.49 times ($82\%$ vs. $33\%$) than the initialization strategy $\text{UNIF}[-10^{-4}, 10^{-4}]$. Additional experiments in Appendix Fig. 27 show that $\text{UNIF}[-10^{-8}, 10^{-8}]$ and $\text{UNIF}[-10^{-16}, 10^{-16}]$ achieve only slightly worse results than $\text{UNIF}[-10^{-12}, 10^{-12}]$. In Appendix F.14, we show a smaller learning rates and a learning rate "warmup" have a different (worse) effect than cold initialization. Results from these ablation experiments encourage us to apply layer normalization, data augmentation, and cold initialization in other experiments.

**Summary.** Our experiments suggest the following design decisions stabilizing contrastive RL: *(i)* Using a simple 3-layer CNN visual feature extractor followed by a wide MLP. *(ii)* Add layer normalization. *(iii)* Initialize last-layer weights of the MLP with small values, $\text{UNIF}[-10^{-12}, 10^{-12}]$. *(iv)* Apply random cropping to both the state and goal images. *(v)* Use a large batch size (2048). We use

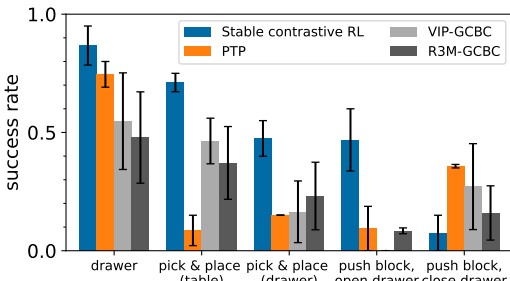

Figure 2: **Evaluation on simulated manipulation tasks.** *(Left)* The five simulated evaluation tasks, with examples of the initial observation (top) and goal observation (bottom). *(Right)* Stable contrastive RL outperforms all baselines on $\frac{4}{5}$ tasks, often by a large margin. The comparison with "contrastive RL" underscores the importance of our design decisions.

**stable contrastive RL** to denote this combination of design decisions and provide a implementation in Appendix E. We call our method "stable" because we find that these design decisions yield more stable learning curves (see Appendix Fig. 13). Additional ablation experiments on `push block, open drawer` (see Appendix Fig. 14) draw similar conclusions.

### 4.3 COMPARING TO PRIOR OFFLINE GOAL-CONDITIONED RL METHODS

Next, we study the efficacy of stable contrastive RL comparing to prior methods, including those that use goal-conditioned supervised learning and those that employ auxiliary representation learning objectives. We run experiments on datasets collected from both simulation and the real-world without any interaction with the environment during training – that is, all experiments focus on offline, goal-conditioned RL.

**Simulation evaluation – manipulation.** We first compare to five baselines that build on goal-conditioned behavioral cloning. The simplest goal-conditioned behavioral cloning ("GCBC") (Chen et al., 2021a; Ding et al., 2019; Emmons et al., 2021; Ghosh et al., 2020; Lynch et al., 2020; Paster et al., 2020; Srivastava et al., 2019) trains a policy to conditionally imitate trajectories reaching goal $g$. Goal-conditioned IQL (GC-IQL) is a goal-conditioned version of IQL (Kostrikov et al., 2021), a state-of-the-art offline RL algorithm. GoFar (Ma et al., 2022a) is an improvement of GCBC that weights the log likelihood of actions using a learned critic function. The fourth baseline is WGCSL (Yang et al., 2022), which augments GCBC with discounted

Figure 3: **Comparing to pretrained representations.** On $\frac{4}{5}$ tasks, stable contrastive RL outperforms baselines that use frozen representations from a VAE (PTP (Fang et al., 2022a)), VIP (Ma et al., 2022b), and R3M (Nair et al., 2022).

advantage weights for policy regression. We also include a comparison with contrastive RL (Eysenbach et al., 2022), which helps us test whether those design decisions improve the performance. Like stable contrastive RL, these baselines are all trained directly on goal images in an end-to-end fashion, not using any explicit feature learning. As shown in Fig. 2, stable contrastive RL matches or surpasses all baselines on four of the five tasks. On those more challenging tasks (`push block, open drawer`; `pick & place (table)`; `pick & place (drawer)`), we see a marked difference between these baselines and stable contrastive RL. However, the baseline methods fail to solve these more challenging tasks. Stable contrastive RL performs worse than GC-IQL and GCBC on one of the tasks (`push block, close drawer`), perhaps because the block in that task occludes the drawer handle and introduces partial observability.

We next compare to three methods that employ learning policy on top of pre-trained representations. PTP (Fang et al., 2022a) focuses on finding sub-goals in a pre-trained VQ-VAE (Van Den Oord et al., 2017) representation space and learns a policy to reach them sequentially. We also compare to a variant of GCBC learning on top of features from VIP (Ma et al., 2022b). We call this method VIP-GCBC. Similarly, our third baseline, R3M-GCBC, trains a policy via supervised learning on top of representations learned by R3M (Nair et al., 2022). Results in Fig. 3 show that stable contrastive RL outperforms all baselines on $\frac{4}{5}$ tasks. While the baselines all require separate objectives for

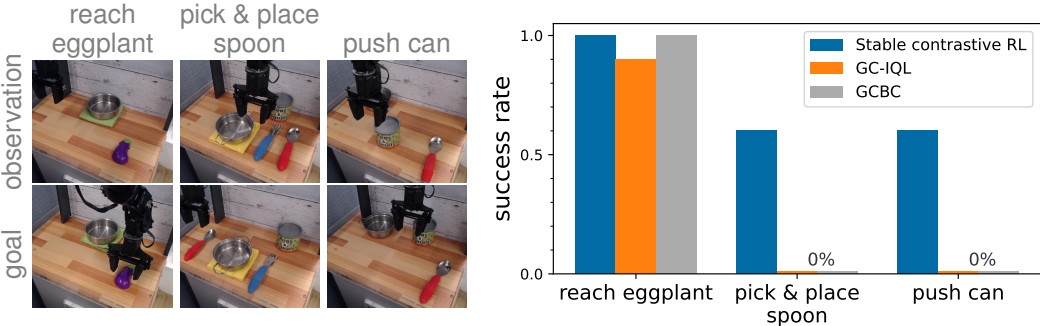

Figure 5: **Evaluation on real manipulation tasks.** Stable contrastive RL matches or outperforms GC-IQL and GCBC on all manipulation tasks. Each success rate is calculated from taking 10 rollouts.

RL and representation learning, stable contrastive RL achieves excellent results without the need for an additional representation learning objective. The full curves can be found in Appendix F.8. Appendices F.11 and F.12 show comparisons against an end-to-end version of PTP and to a variant without sub-goal planning.

**Simulation evaluation – locomotion.** Our next set of experiments focus on locomotion tasks. We conduct these experiments to see whether the same design decisions are helpful on tasks beyond manipulation, and on tasks from which we have only suboptimal data. We compare stable contrastive RL against contrastive RL, GCBC, and GC-IQL on two goal-reaching tasks from prior work (Mendonca et al., 2021): `Walker` and `Quadruped`. We evaluate all methods in the offline setting, recording the average success rates over reaching 12 unique goal poses. As shown in Fig. 4, stable contrastive RL outperforms all three baselines on both locomotion tasks, achieving a 33% higher (relative) average success rate than the strongest baseline.

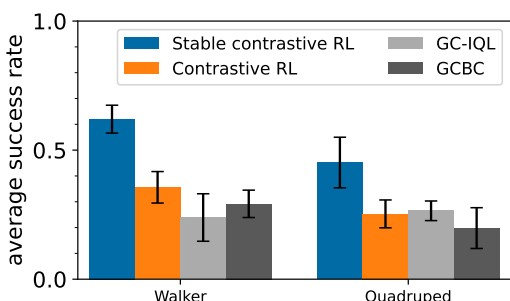

Figure 4: **Evaluation on simulated locomotion tasks.** Stable contrastive RL outperforms contrastive RL, GC-IQL and GCBC in two locomotion tasks. Each success rate is calculated by averaging over 12 diverse goal poses (Mendonca et al., 2021).

**Real-world evaluation.** We next study whether these methods can effectively solve real world, goal-directed, image-based manipulation tasks, learning entirely from offline data. We compare stable constrastive RL against the strongest performing baseline from the simulated results, GCBC. We compare to GC-IQL as well because it performed well in the simulated experiments, and its simplicity makes it practically appealing. We evaluated on three goal-reaching tasks and report success rates in Fig. 5. Stable contrastive RL performs similarly to baselines on the simple (`reach eggplant`) task, while achieving a 60% success rate on the two harder tasks (`pick & place spoon`, `push can`), where all baselines fail to make progress. We conjecture that this good performance might be explained by the hard-negative mining dynamics of stable contrastive RL, which are unlocked by our design decisions.

In Sec. 4.4, we visualize the representations learned by stable contrastive RL and baselines, providing an intuition for why our method achieves higher success rates. Sec. 4.5 and Appendix F.7 include comparisons of Q values learned by our method and baselines at different time steps of a trajectory, aiming to quantitatively analyze the advantage of our method.

## 4.4 Visualization of Learned Representations

To understand why stable contrastive RL achieved high success rates in Sec. 4.3, we study whether the representations acquired by self-supervised RL contain task-specific information. To answer this question, we visualize the representations learned by stable contrastive RL, a VAE, and GCBC on the `push block, open drawer` task. Given an initial image (Fig. 6 *far left*) and the desired goal image (Fig. 6 *far right*), we interpolate between the representations of these two images, and retrieve the nearest neighbor in a held-out validation set. For comparison, we also include direct interpolation in pixel space. As expected, interpolating in pixel space generates unrealistic images. While linearly interpolating in the VAE and GCBC representation spaces produce realistic images, the retrieved images fails to capture causal relationships, motivating the

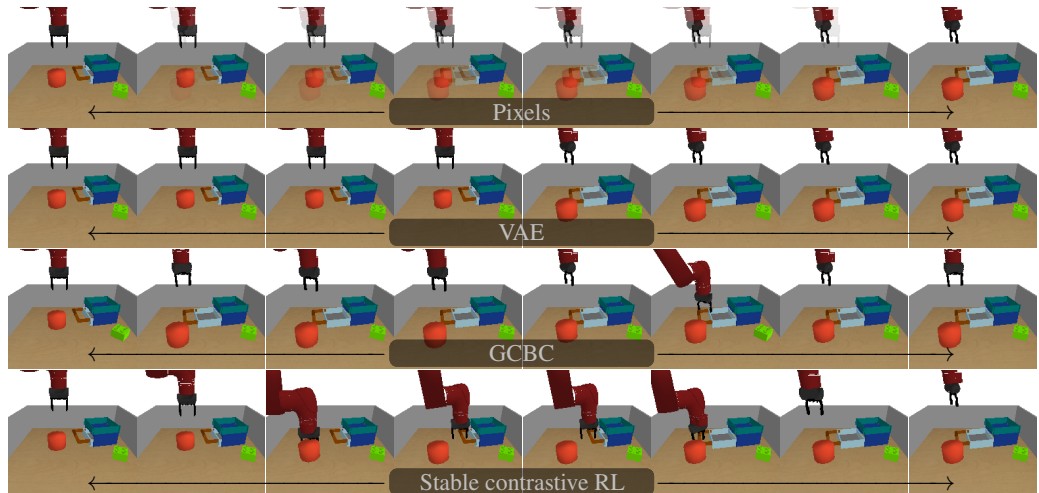

Figure 6: **Visualizing the representations.** *(Row 1)* Directly interpolating between two images in pixel space results in unrealistic images. *(Row 2)* Using a VAE, we interpolate between the representations of the left-most and right-most images, visualizing the nearest-neighbor retrievals from a validation set. The VAE captures the contents of the images but not the causal relationships – the object moves without being touched by the robot arm. *(Row 3)* GCBC also produces realistic images while ignoring temporal causality. *(Row 4)* Stable contrastive RL learns representations that capture not only the content of the images, but also the causal relationships – the arm first moves *away* from its position in the goal state so that it can move the object into place.

necessity of adding other machinery, e.g., relabeling latent goal (Nair et al., 2018; Pong et al., 2020; Rosete-Beas et al., 2022) and latent sub-goal planning with value constraints (Fang et al., 2022a;b). When interpolating the representations of stable contrastive RL, the intermediate representations correspond to sequence of observations that the policy should visit when transiting between the initial observation and the final goal. These results suggest that stable contrastive RL acquires causal relationships in the learned representations, providing an intuition for its good performance. Appendices F.5 and F.6 contain a quantitative experiment and additional visualizations.

## 4.5 THE ARM MATCHING PROBLEM

Previous works have found that some goals are easier to be distinguish from the others (easy negative examples) (Rosete-Beas et al., 2022; Tian et al., 2020; Alakuijala et al., 2022), while some goals might require temporal-extended reasoning to reach (hard negative examples). For example, on task `push block`, `open drawer`, prior goal-conditioned algorithms (Fang et al., 2022a; Kostrikov et al., 2021) simply match the arm position with the one in the goal image and fail to move the green block to its target position. We call this failure *the arm matching problem*. Stable contrastive RL learns the critic using contrastive loss (Eq. 2) and might perform sort of hard negative mining, assigning Q values correctly and avoiding the arm matching problem.

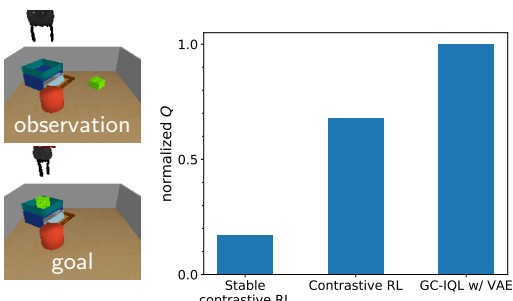

Figure 7: **Misclassifying success.** GC-IQL mistakenly predicts that this observation has succeeded in reaching this goal, even though the object is in the incorrect position. In contrast, stable contrastive RL recognizes that this observation is far from the goal and assigns it a low Q value.

We test this hypothesis in Fig. 7 by comparing the Q values learned by stable contrastive RL and contrastive RL, aiming to assess whether our design decisions lead to more accurate estimates of the Q-values. Prior work has suggested that pre-trained VAE representations can also be effective, so we also compare to a version of GC-IQL applied on top of VAE features (Fang et al., 2022a;b). We normalize these Q values by the minimum and maximum values in a rollout. Fig. 7 compares the Q values for an observation where the object is not in the correct location, but the gripper is in the correct position. GC-IQL (mistakenly) assigns high Q values to this observation. In contrast, stable contrastive RL predicts a small value for this observation, highlighting the accuracy of its Q function and showing how it can avoid arm matching behavior and achieves higher success rates. Appendix F.7 includes a comparison of Q values in an optimal trajectory.

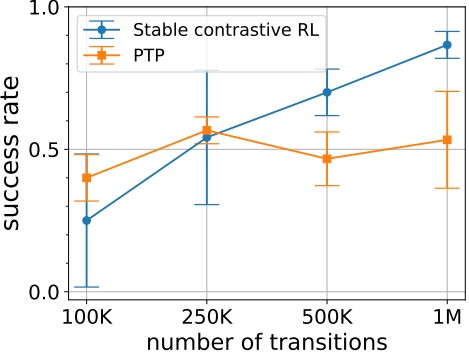

Figure 8: **The influence of dataset size.** Increasing the amount of offline data boosts the success rate of stable contrastive RL, while the performance of PTP saturates after 250,000 transitions.

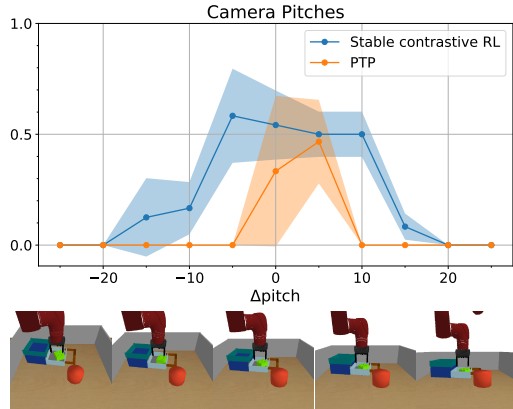

Figure 9: **Stable contrastive RL ceneralizing to unseen camera angles**, as well as variations in yaw and object color (see Appendix Fig. 23).

### 4.6 THE INFLUENCE OF DATASET SIZE ON PERFORMANCE

Scaling model performances with the amount of data have been successfully demonstrated in CV and NLP (Brown et al., 2020; He et al., 2022), motivating us to study whether contrastive RL offers similar scaling capabilities in the offline RL setting. To answer this question, we run experiments with dataset sizes increasing from 100K to 1M, comparing stable contrastive RL against a baseline that uses pre-trained features to improve policy learning. Results in Fig. 8 shows that stable contrastive RL effectively makes use of additional data, with a success rate that increases by $3\times$ as we increase the dataset size. In contrast, the performance of PTP saturates around 250K transitions, suggesting that most of the gains observed for stable contrastive RL may be coming from better representations, rather than from a better reactive policy. In Appendix F.9 we find that increasing the dataset size also improves the binary accuracy.

### 4.7 GENERALIZING TO UNSEEN CAMERA ANGLES AND OBJECT COLORS

Our next set of experiments study the robustness of the policy learned by stable contrastive RL. We hypothesize that stable contrastive RL might generalize to unseen tasks reasonably well for two reasons: *(1)* stable contrastive RL resembles the contrastive auxiliary objectives used by prior work to improve robustness (Nair et al., 2022; Laskin et al., 2020b;a); *(2)* because stable contrastive RL learns features solely with the critic objective, we expect that the features will not retain task-irrelevant information (unlike, say, representation based on auto-encoding). We ran an experiment comparing the generalization of stable contrastive RL against a baseline that uses features pre-trained via reconstruction by varying the environment. We show results on varying pitch in Fig. 9, and include the other results and full details in Appendix F.10. These experiments provide preliminary evidence that the policy learned by our method might generalize reasonably well. In Appendix F.11 & F.12 we study the effect of augmenting stable contrastive RL with auxiliary objectives and sub-goal planning, which are important for prior work (Fang et al., 2022a) but decrease performance of stable contrastive RL.

## 5 CONCLUSION

In this paper, we have studied design decisions that enabled a self-supervised RL method to solve real-world robotic manipulation tasks that stymie prior methods. We have found that decisions regarding the architecture, batch size, normalization, initialization, and augmentation are all important.

**Limitations.** Much of the motivation behind goal-conditioned RL is that it may enable the same sorts of scaling laws seen in self-supervised methods for computer vision (Zhai et al., 2022) and NLP (Kaplan et al., 2020). Our work is only a small step in developing self-supervised RL methods that work on the real-world robotics datasets; relative to the model sizes and dataset sizes in NLP and CV, ours are tiny. However, our design decisions do allow us to significantly improve performance in the offline setting (Fig. 1) and do yield a method whose success rate seems to increase linearly as the dataset size doubles (Appendix Fig. 22). So, we are optimistic that these design decisions may provide helpful guidance on further scaling these methods.

**Acknowledgements.** We thank Ravi Tej for discussions about this work, and anonymous reviewers for providing feedback on early versions of this work. We thank Seohong Park for finding issues in the environment implementation. BE is supported by the Fannie and John Hertz Foundation and the NSF GRFP (DGE2140739). RS is supported in part by ONR N000142312368 and DARPA/AFRL FA87502321015. HW, KF, and SL are supported in part by ONR N000142012383, and KF is also supported by a CRA CIFellows Award.

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

## A    COMPARISON OF RELATED WORK

We compare prior works focusing on goal-conditioned RL, representation learning, and model learning in Table 1.

Table 1: **Self-supervised RL from offline data.**

|  | Reward-free | Avoids Predicting Pixels | Directly produces a policy |
|---|---|---|---|
| Contrastive RL | ✓ | ✓ | ✓ |
| Representation Learning | ✓ | ✓ / ✗ | ✗ |
| Model Learning | ✓ / ✗ | ✗ | ✗ |

## B    REVIEW OF TD CONTRASTIVE RL

The recursive definition of the discounted state occupancy measure (Eq. 1) allows us to rewrite Eq. 2 into a *temporal different* (TD) variant (Eysenbach et al., 2020), with $w(s, a, s_f) \triangleq \frac{\sigma(f(s,a,s_f))}{1-\sigma(f(s,a,s_f))}$,

$$\max_f \mathbb{E}_{\substack{(s,a)\sim p(s,a),s'\sim p(s'|s,a) \\ s_f\sim p(s_{t+}),a'\sim\pi(a'|s,g)}} \Big[(1-\gamma)\log\sigma(f(s,a,s')) \tag{3}$$

$$+ \log(1-\sigma(f(s,a,s_f))) + \gamma\lfloor w(s',a',s_f)\rfloor_{\text{sg}}\log\sigma(f(s,a,s_f))\Big].$$

A prototypical example of this TD variant is C-Learning (Eysenbach et al., 2020) that lies in the family of contrastive RL methods (Eysenbach et al., 2022). We will focus on the TD learning critic objective in our discussion and adapt it to the offline setting with a goal-conditioned behavioral cloning regularized policy objective:

$$\max_{\pi(\cdot|\cdot,\cdot)} \mathbb{E}_{p_g(g)p(s,a_{\text{orig}})\pi(a|s,g)} \left[(1-\lambda)\cdot f(s,a,g) + \lambda\log\pi(a_{\text{orig}} \mid s,a)\right]. \tag{4}$$

## C    INTUITION: A CONNECTION BETWEEN CONTRASTIVE RL AND HARD NEGATIVE MINING

One practical challenges with learning value functions or distance functions for goal-reaching tasks is that visually similar images may actually require a large number of steps to transit between. Prior work has addressed this challenging by manually mining hard negative examples (Tian et al., 2020; Rosete-Beas et al., 2022). Our motivation for studying contrastive methods was a hypothesis that, with appropriate design decisions, contrastive methods would automatically induce a form of hard-negative mining. In particular, we wrote the gradient of the objective (Eq. 2). with respect to the representation of random goals $\psi(s_f^-)$,

$$\frac{\partial}{\partial\psi(s_f^-)}\mathcal{L}_2(\phi(s,a),\psi(s_f^-)) = -\sigma(\phi(s,a)^\top\psi(s_f^-))\phi(s,a).$$

This gradient "pushes" the representation of a random goal $\psi(s_f^-)$ away from the anchor $\phi(s,a)$ with a larger weight when it is misclassified. Intuitively, if we think of these weights as importance weights, then this gradient corresponds to finding the hardest negative examples within the batch. We conjecture that realizing this effect will require a large batch size, as well as a few other design decisions, which we discuss next.

## D    ARCHITECTURE DIAGRAM

An overview of our network architecture is shown in Fig. 10.

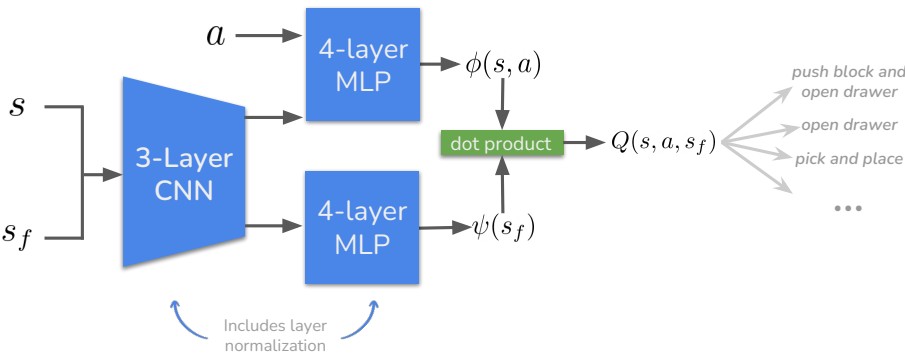

Figure 10: **Overview of network architecture.** Our experiments suggest that this architecture can stabilize and accelerate learning for contrastive RL. Layer normalization is applied to each layer of the visual feature extractor (CNN) and contrastive representations (MLP). We initialize weights in final layers of MLP close to zero to encourage alignment of contrastive representations, and use data augmentation to prevent overfitting.

## E   EXPERIMENTAL DETAILS

**Tasks.**   Our experiments use a suite of simulated and real-world goal-conditioned control tasks based on prior work (Fang et al., 2022a;b; Ebert et al., 2021; Mendonca et al., 2021). *First*, we use a benchmark of five manipulation tasks proposed in  (Fang et al., 2022a;b) (Fig. 2). The observations and goals are $48 \times 48 \times 3$ RGB images of the current scene. These tasks are challenging benchmarks because solving most of them requires multi-stage reasoning. *Second*, we use the goal-conditioned locomotion benchmark proposed in (Mendonca et al., 2021). The observations and goal poses are $64 \times 64 \times 3$ RGB images. This benchmark helps study those design decisions for offline goal-reaching beyond manipulation. *Third*, we study the performance of our algorithm in real-world using a 6-DOF WidowX250 robot arm. We use Bridge dataset (Ebert et al., 2021) where observation and the goal are $128 \times 128 \times 3$ RGB images with much noisier backgrounds than the simulated tasks. For evaluation, we use three tasks from `Toy Kitchen 2` scene in the dataset where the initial and desired object positions are unseen during training (see Fig. 5). These tasks are challenging because they require multi-stage reasoning (e.g., the drawer can only be opened after the orange object has been moved).

**Offline dataset.**   The offline manipulation dataset we used in simulation consists of near-optimal demonstrations of primitive behaviors, such as opening the drawer, pushing blocks, and picking up objects. The scripted data collection policy can access the underlying object states. The lengths of demonstration vary from 50 to 100 time steps and the offline dataset approximately contains 250K transitions in total. We note that none of the offline trajectories complete the demonstration from the initial state to the goal. For evaluation, we create a dataset of 50 goals by randomly sampling the positions of objects and the robot arm, and evalaute the success rate of reaching these goals. The locomotion benchmark includes a mixture of optimal and suboptimal transitions. We randomly collect 100K transitions from the replay buffer of a policy trained with LEXA (Mendonca et al., 2021), which achieves success rates of $75\%$ on `Walker` and $58\%$ on `Quadruped` at convergence. Our real-world offline dataset uses the existing Bridge Data (Ebert et al., 2021), including around 20K demonstrations of various manipulation skills. Note that some trajectories in this dataset are suboptimal since they were collected by a weakly scripted data collection policy that sometimes misses grasping the target objects. This dataset provides broad and diverse demonstrations to evaluate whether an agent is able to stitch skills and generalize to different scenes.

**Implementations.**   We implement stable contrastive RL using PyTorch (Paszke et al., 2019)[2]. We use the open-sourced implementation of PTP[3] as our baseline and adapt the underlaying IQL algorithm to GC-IQL and GCBC. Unless otherwise mentioned, we use same hyperparameters as this

---

[2]https://anonymous.4open.science/r/stable_contrastive_rl-5A42.

[3]https://github.com/patrickhaoy/ptp

Table 2: Hyperparameters for stable contrastive RL.

| Hyperparameters | Values | Notes |
|---|---|---|
| batch size | 2048 | |
| number of training epochs | 300 | |
| number of training iterations per epoch | 1000 | equivalent to number of gradient steps per epoch |
| dataset size | 250K | number of transitions $(s, a, s', s_f)$ |
| image size | $48 \times 48 \times 3$; $128 \times 128 \times 3$ | Size of RGB images in the simulated and real-world tasks |
| episode length | 400 | |
| image encoder architecture | 3-layer CNN | kernel size = (8, 4, 3), number of channels = (32, 64, 64), strides = (4, 2, 1), paddings = (2, 1, 1) |
| policy network architecture | (1024, 4) MLP | |
| critic network architecture | (1024, 4) MLP | |
| weight initialization for final layers of critic and policy | $\text{UNIF}[-10^{-12}, 10^{-12}]$ | |
| policy stand deviation | 0.15 | |
| contrastive representation dimension | 16 | |
| data augmentation | random cropping | replicating edge pixel-padding size = 4 |
| augmentation probability | 0.5 | applying random cropping to both $s$ and $g$ in the GCBC regularizer in Eq. 4 with this probability |
| discount factor | 0.99 | |
| learning rate of Adam (Kingma & Ba, 2015) optimizer | 0.0003 | |

implementation. For baselines WGCSL[4], GoFar[5], VIP-GCBC[6], and R3M-GCBC[7], we adapted the implementation provided by the authors.

**Hyperparameters and computation resources.** We implement stable contrastive RL using PyTorch again on top of the public PTP codebase. Our algorithm is adapted from contrastive RL implementation[8] and incorporate those novel design decisions mentioned in Sec. 4.2. We summarize hyperparameters in Table 2. For each experiment, we allocated 1 NVIDIA V100 GPU and 64 GB of memories to do computation.

---

[4]https://github.com/YangRui2015/AWGCSL
[5]https://github.com/JasonMa2016/GoFAR
[6]https://github.com/facebookresearch/vip
[7]https://github.com/facebookresearch/r3m
[8]https://github.com/google-research/google-research/tree/master/contrastive_rl

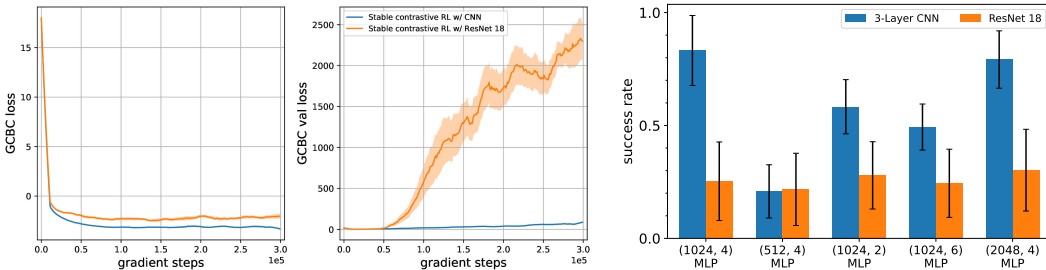

Figure 11: ResNet 18 is susceptible to overfitting for the policy on offline datasets.

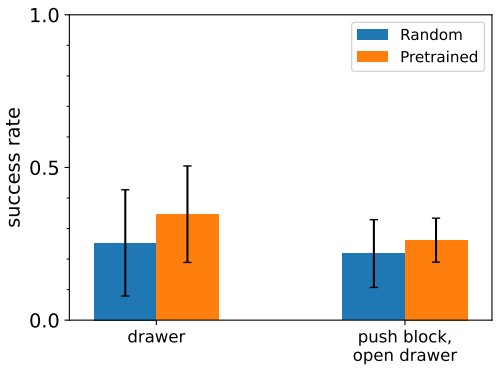

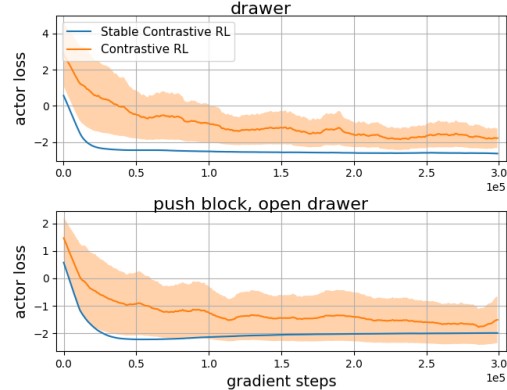

Figure 12: Using weights pretrained on supervised learning tasks might boosts performances of stable contrastive RL.

Figure 13: "Stable" contrastive RL is more stable than the original contrastive RL.

# F ADDITIONAL EXPERIMENTS

## F.1 OVERFITTING OF THE RESNET 18 ENCODER

We hypothesize that the over-parameterized ResNet 18 visual encoder overfits to the dataset when comparing with the 3-layer CNN. To study this, we conduct experiments on task `push block, open drawer`, showing training and validation loss of the GCBC regularization for both visual backbones. Note that we apply the cold initialization, layer normalization, and random cropping data augmentation introduced in Sec. 4.2 to both ResNet 18 and 3-layer CNN to make a fair comparison in Fig. 1 and these additional experiments. The observation that the training loss remains low while the validation loss starts to increase after 50K gradient steps (Fig. 11 *(Left)*) suggests that the larger ResNet 18 is susceptible to overfitting for the policy. Additionally, we include ablations between 3-layer CNN and ResNet 18 with different subsequent MLP architectures for completeness in Fig. 11 *(Right)*. We find that ResNet 18 with deeper and wider MLP architecture does not improve the performance significantly and the best combination still underperforms Stable contrastive RL by $2.75\times$. Together with the GCBC training and validation losses above, we suspect that the ResNet 18 overfits to the training data *despite* the data augmentation (random cropping) we applied in attempts to mitigate this.

## F.2 EFFECT OF PRETRAINING

We ran additional experiments to study the effect of pretrained models for contrastive RL. We applied our design decisions to stable contrastive RL variants that use a ResNet-18 visual encoder and compared two initialization strategies: initialization with random weights and initialization with weights pretrained on ImageNet (He et al., 2016). In Fig. 12, we report results on `drawer` and `push block, open drawer`, a challenging multi-stage task that involves first pushing the block and then opening the drawer.

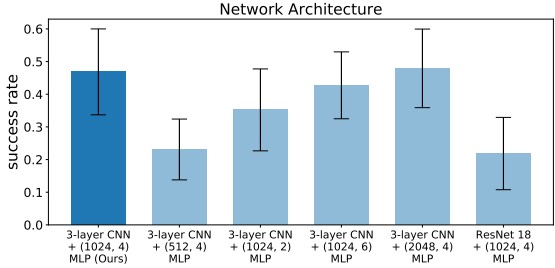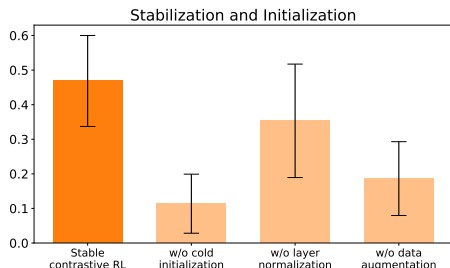

Figure 14: Additional ablations on `push block, open drawer`.

We find that using pretrained weights boosts performance of stable contrastive RL on the relatively simple `drawer` but does not significantly benefit the algorithm on `push block, open drawer`. These results suggest that those weights pretrained on supervised learning tasks might help the learning of contrastive RL in some cases. However, we note that we have only explored one pretraining method. Different pretraining datasets and methods (e.g., R3M (Nair et al., 2022), VIP (Ma et al., 2022b)) may produce different results. Nonetheless, we leave the investigation of which pretraining methods best accelerate stable contrastive RL to future work.

### F.3 Understanding "Stable" in Stable Contrastive RL

The reason for using "stable" contrastive RL to name our method is that the original contrastive RL (Eysenbach et al., 2022) is unstable. We provide a comparison between the learning curves of stable contrastive RL and contrastive RL on tasks `drawer` and `push block, open drawer`. The observation that the actor loss of stable contrastive RL are less oscillatory than those of the original contrastive RL demonstrates that our design decisions improve stability.

### F.4 Additional Ablations

Our ablation experiments are done on two tasks: `drawer` and `push block, open drawer`, both using image-based observations. There are three reasons to select these two tasks. First, while using state-based tasks would have been computationally less expensive, we opted for the image-based versions of the tasks because they more closely mirror the real-world use case (Fig. 5), where the robot has image-based observations. Second, prior methods struggle to solve both these tasks, including (original) contrastive RL (Fig. 2). Thus, there was ample room for improvement. Third, out of the tasks in Fig. 2, we choose the easiest task (`drawer`, which is still challenging for baselines) and one of the most complex tasks (`push block, drawer open`). These two representative tasks are a good faith effort to evaluate our design decisions.

Additional ablations compare stable contrastive RL with the same set of design decision variants as in Fig. 1 on tasks `push block, open drawer`. We report the results in Fig. 14. Regarding network architectures, we find that an MLP with 2, 4, and 6 numbers of layers and 1024 units perform similarly on this task, suggesting that our choice, an (1024, 4) MLP, is still effective. For stabilization and initialization techniques, we find that cold initialization and data augmentation still improve the performance significantly, while removing layer normalization decreases the performance by $24.5\%$. Taken together, our design decisions continue to outperform or perform similarly to other variants on this new task.

### F.5 More Examples of Latent Interpolation

We show more examples of interpolating different representations on tasks `drawer`, `pick & place (table)`, and `pick & place (drawer)` in Fig. 15, Fig. 16, and Fig. 17 respectively.

### F.6 Measuring Interpolation

To quantitatively study interpolation, we run another set of experiments comparing the similarity of interpolated representations (images) with the ground truth sequence on `push block, open drawer`. Specifically, we retrieve the nearest neighbor of each interpolated representation from a policy rollout and label the time step of interpolations accordingly (perm[$t$]).

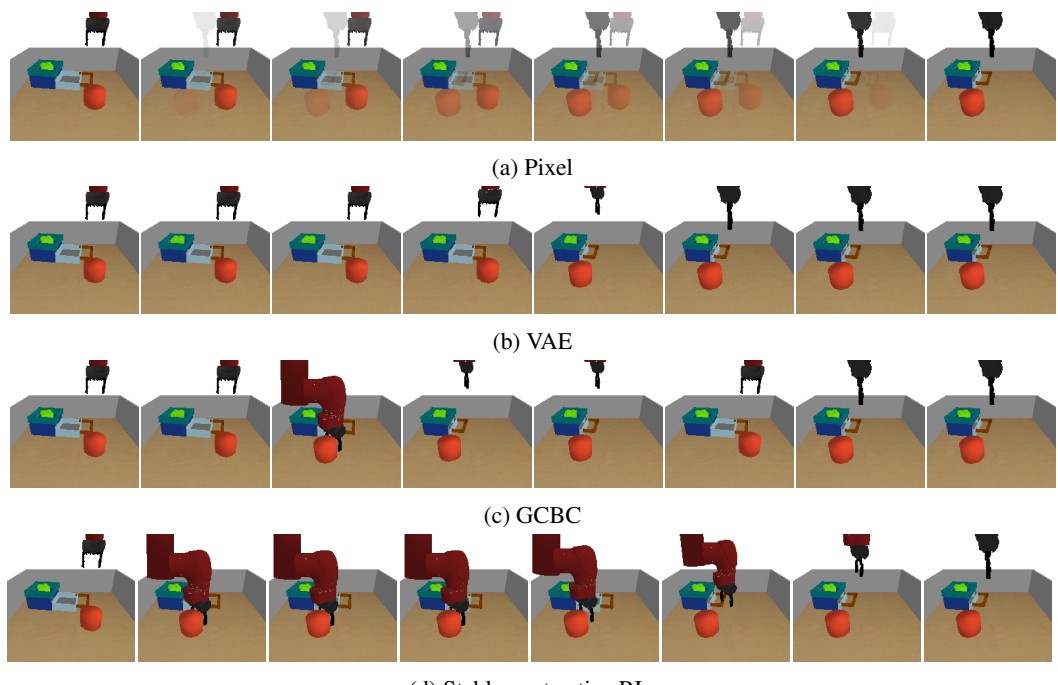

(a) Pixel

(b) VAE

(c) GCBC

(d) Stable contrastive RL

Figure 15: Interpolation of different representations on `drawer`.

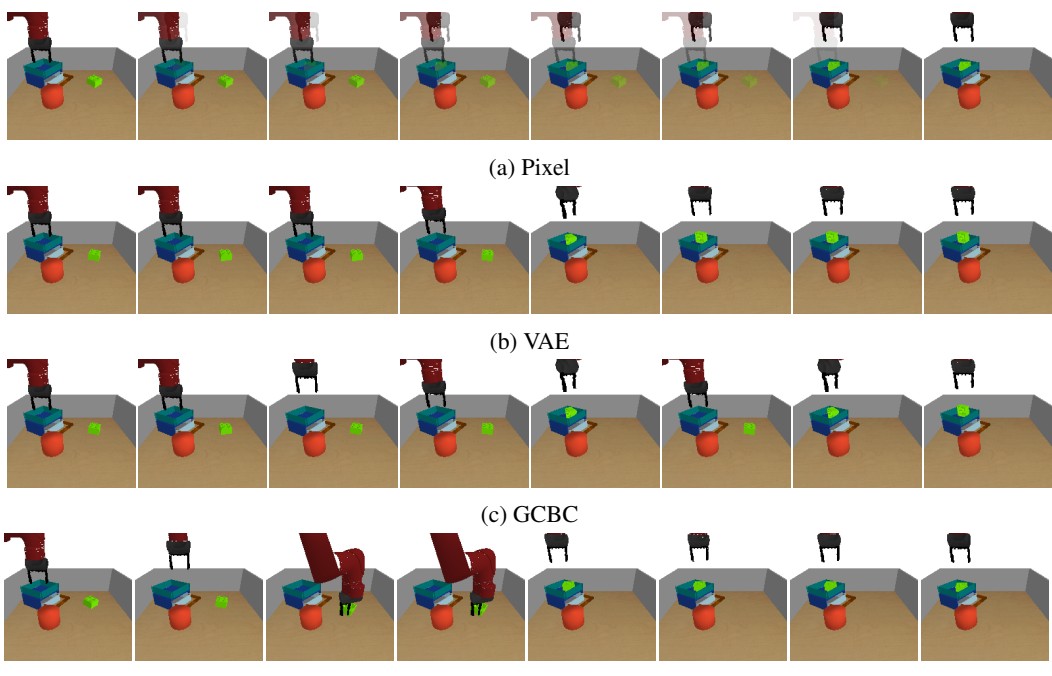

(a) Pixel

(b) VAE

(c) GCBC

(d) Stable contrastive RL

Figure 16: Interpolation of different representations on `pick & place (table)`.

With these time steps, we define a permutation error with respect to the ground truth time steps:

$$\sum_{t=0}^{T} |\text{perm}[t] - t|$$

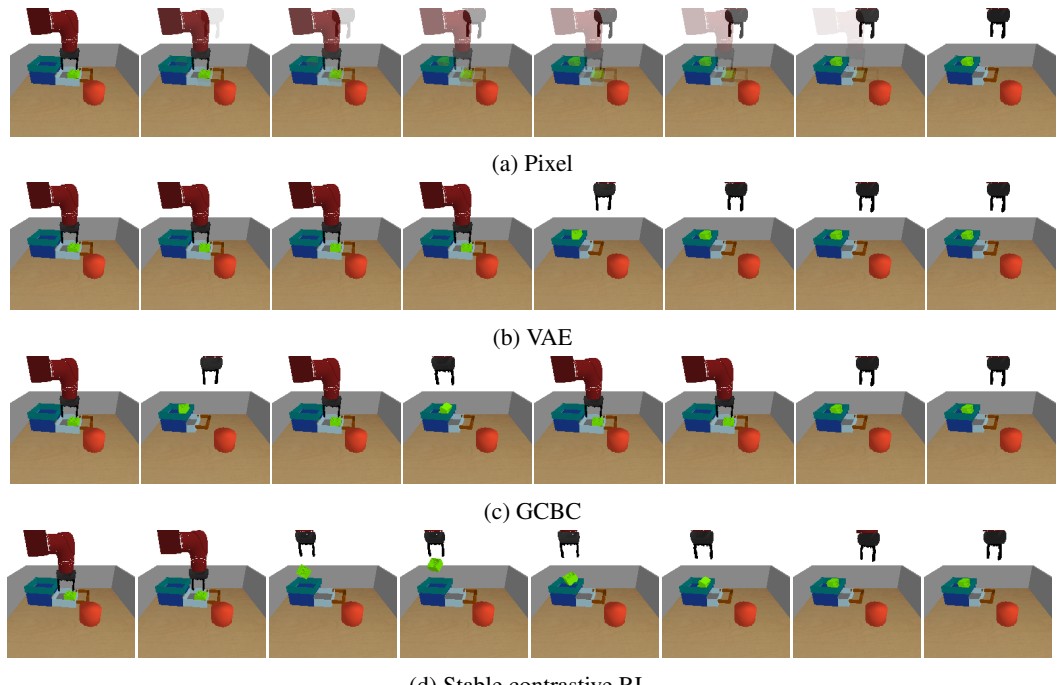

(a) Pixel

(b) VAE

(c) GCBC

(d) Stable contrastive RL

Figure 17: Interpolation of different representations on `pick & place (drawer)`.

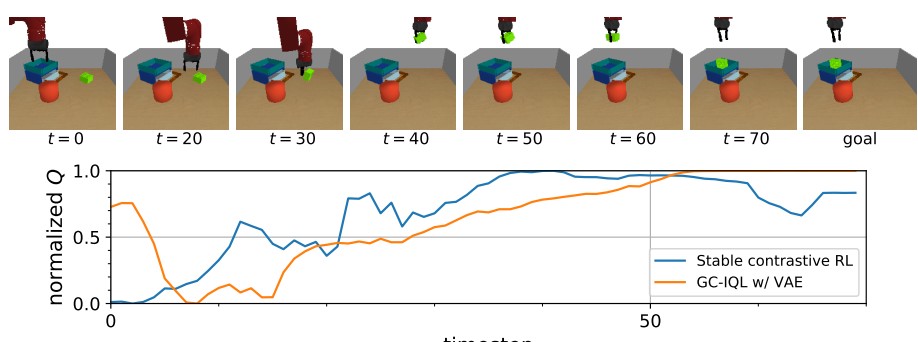

Figure 19: **Predictions of success.** Stable contrastive RL learns accurate Q values for the task of reaching the goal shown above: they increase throughout a successful rollout. On the contrary, the GC-IQL baseline mistakenly predicts that picking up the green object *decreases* the Q-value, likely because the GC-IQL representations ignore the position of the green block. This experiment suggests that representations derived from a VAE, like those learned by GC-IQL and many prior methods (Yarats et al., 2019; Nair et al., 2018; Nasiriany et al., 2019), may be less effective at predicting success than those learned by contrastive RL methods.

Results in Fig. 18 demonstrate that interpolation in the VAE representation space and the pixel space are not well-aligned with the ground truth time steps (better than random permutations), while contrastive representations achieves a lower error, suggesting that it might contain information that is uniquely well-suited for control and potentially leverage a goal-conditioned policy.

## F.7 ARM MATCHING

To evaluate the performance of contrastive RL in tackling the arm matching problem, we collect a trajectory of contrastive RL trained on the offline dataset and comparing the Q value predicted by stable contrastive RL with a GC-IQL trained on top of VAE representations, assuming that pre-trained features might help mitigate the arm matching problem as well. We normalize both Qs by values of the minimum and the goal. As shown in Fig. 19, stable contrastive RL correctly learns a monotonic increasing Q throughout the rollout, while GC-IQL learns a V-shape Q relating higher values to a

closer arm to the target position, ignoring the position of the green block ($t = 30$). This experiment demonstrates that representations derived from VAE may be less efficient at predicting success than contrastive representations in the scene involving temporal-extended reasoning.

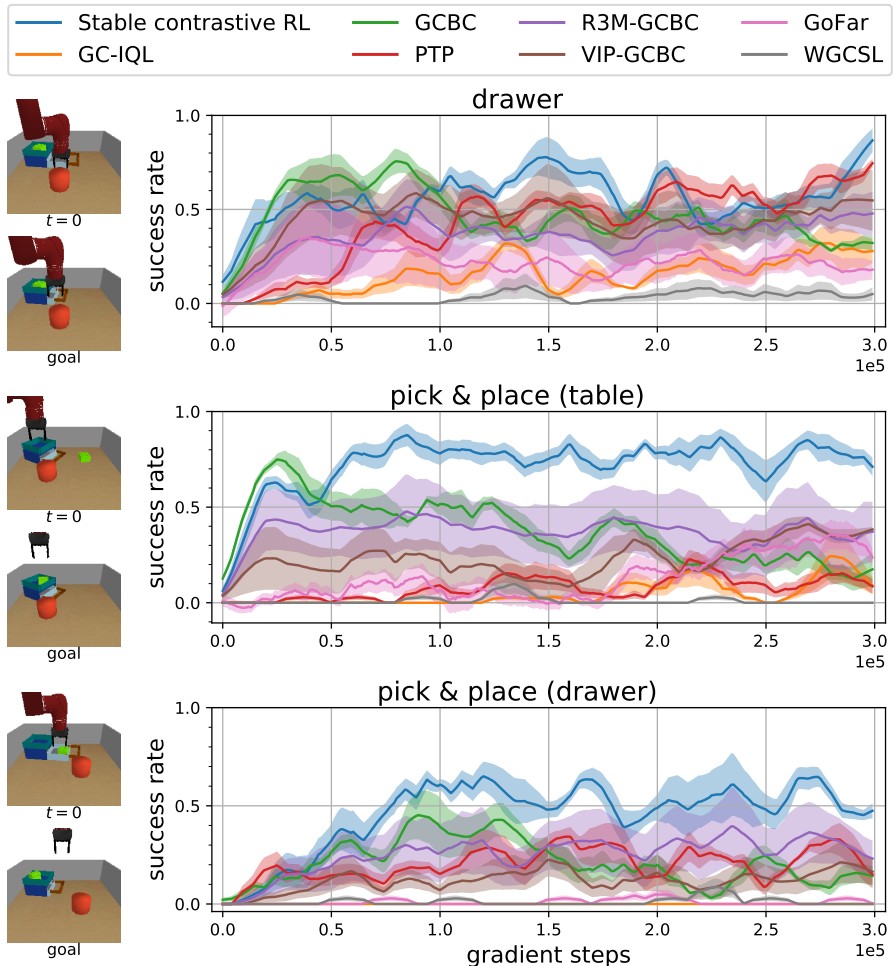

Figure 20: **Evaluation on simulated manipulation tasks**. Stable contrastive RL outperforms all baselines on `drawer`, `pick & place (table)`, and `pick & place (drawer)`.

## F.8 EVALUATION ON SIMULATED MANIPUATION TASKS

We report results in Fig. 20 and Fig. 21 with curves indicating mean success rate and shaded regions indicating standard deviation across 5 random seeds after 300K gradient steps' training. Stable contrastive RL outperforms or achieves similar performance on 4 out of 5 tasks comparing to other baselines. These results suggest that when accompanied with proper techniques, stable contrastive RL is able to leverage a diverse offline dataset and emerges a general-purpose goal-conditioned policy, thus serving as a competitive offline goal-conditioned RL algorithm.

## F.9 DATASET SIZE

Scaling model performances with the amount of data have been successfully demonstrated in CV and NLP (Brown et al., 2020; He et al., 2022), motivating us to study whether contrastive RL offers similar scaling capabilities in the offline RL setting. Sec. 3.2 have discussed the benefits of training with more positive and negative pairs to contrastive representation learning, suggesting that increasing amounts of data might also boost the performance of contrastive RL. We empirically study the

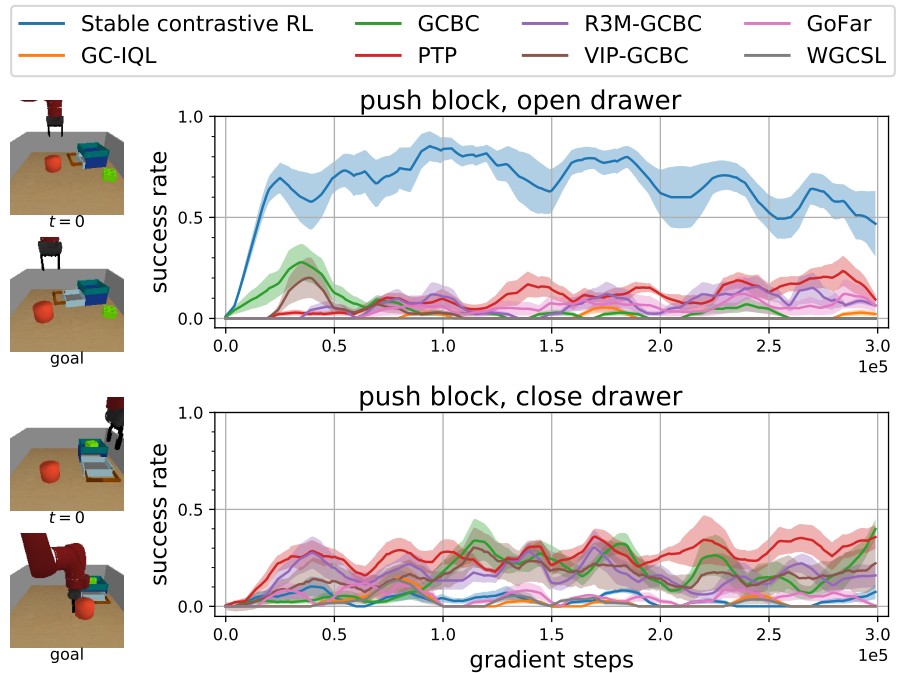

Figure 21: **Evaluation on simulated manipulation tasks**. Stable contrastive RL outperforms or performs similarly to all baselines on `push block, open drawer` and `push block, close drawer`.

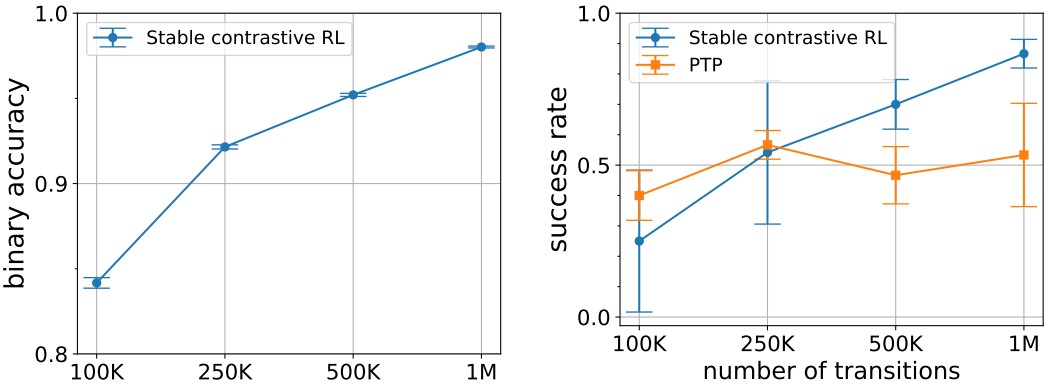

Figure 22: Increasing the amount of offline data improves the binary future prediction accuracy of stable contrastive RL and boosts the success rate.

hypothesis that *stable contrastive RL learns a representation that achieves better performances with growing dataset size.*

To test this hypothesis, we create datasets of different sizes containing various manipulation skills, select task `drawer` for evaluation, and compare the accuracy of stable contrastive RL in predicting future states on each of the datasets. We compare stable contrastive RL against PTP that is based on a pre-trained VAE. We compare both algorithms by training three random seeds of them for 300K gradient steps on each dataset. Fig. 22 shows the mean and standard deviation of success rates over 10 rollouts and binary future prediction accuracies computed between critic outputs and ground truth labels. Observe that when the size of dataset *increased*, the binary accuracy of contrastive RL *increase* accordingly ($84\% \to 98\%$), suggesting that the algorithm indeed strengthens its representation in future prediction. These results make sense, as increasing the size of the dataset is especially important for contrastive approaches that directly use all of the data as *negative* examples. In addition, our experiments show that contrastive RL consistently improves its success rates when we use larger dataset: $31\% \to 84\%$, verifying our hypothesis. We suspect that the emerging of a denser distribution

of positive and negative pairs given a larger dataset might facilitate policy learning. In contrast, a steady or even lower success rate of PTP given the increasing amounts of data might due to the capacity limitation of pre-trained VAE representation.

## F.10 GENERALIZATION

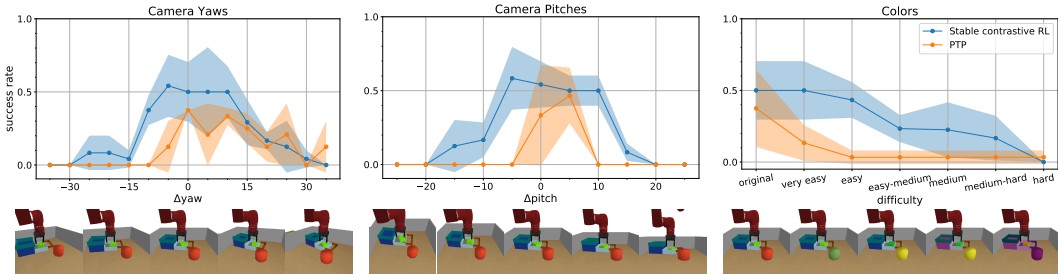

Figure 23: **Generalizing to unseen tasks.** Stable contrastive RL retrains good performance on a broad range of unseen camera angles (yaws / pitches) and object colors.

One of the drawback of driving RL by pre-trained visual representation learning is tying algorithm performance with the latent representation to reconstruct corresponding images. Figure 6 has shown the *support* of VAE representation distribution could be discontinuous and there could be noise in a out-of-distribution reconstructed image, indicating the degradation of RL performance on unseen tasks. Rather than predicating on a pre-trained visual representation, stable contrastive RL trains its self-supervised representation with the actor-critic framework (Eysenbach et al., 2022), expecting to acquire a more robust and generalizable algorithm. We hypothesize that *stable contrastive RL generalizes better to unseen tasks than prior method*.

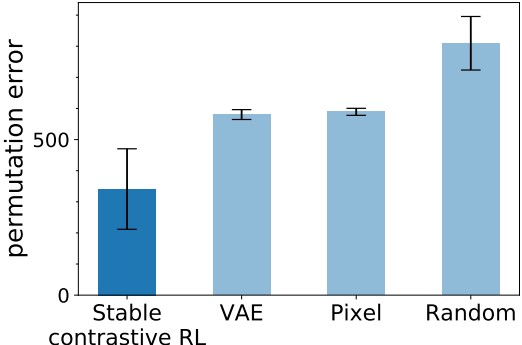

Figure 18: To quantitatively evaluate interpolation, we measure the similarity of the retrieved representations with the ground truth sequence. Stable contrastive RL achieves lower error than the alternative methods. See text for details.

To test this hypothesis, we ran experiments with one of our goal-conditioned tasks in three visually different scenarios (Fig. 23): varying the camera yaw angle, varying the camera pitch angle, and varying the color of objects in the scene. We evaluate the policy of stable contrastive RL and PTP after 300K gradient steps' offline training in these three scenerios, plotting the mean and standard deviation of success rate across three random seeds. Since the color of objects cannot be quantified precisely, we change colors of one to several objects and define the difficulty of each scene accordingly. As shown in Fig. 23, PTP is highly sensitive to both camera viewpoints and object colors: large values of camera $\Delta$yaw and $\Delta$pitch and significant different object colors result in uninformative VAE representation, PTP consequently performs poorly. Stable contrastive RL, which does not depend on VAE representation, outperforms PTP and is robust to a wide range of camera angles and object colors. These experiments provide preliminary evidence that contrastive representations might generalize reasonably well.

## F.11 ADDITIONAL PERCEPTUAL LOSSES

Prior work has found that auxiliary visual loss is necessary and important to drive an RL algorithm (Yarats et al., 2019). However, visualization and analysis in Fig. 6 suggest that stable contrastive RL is already acquiring useful representations, leading us to hypothesize that it may not require additional perception-specific losses. To test this hypothesis, we compare contrastive RL (with representations learned end-to-end) with a variant that uses a pre-trained VAE representation as the input for both actor and critic. We compare against PTP (which uses the same VAE representa-

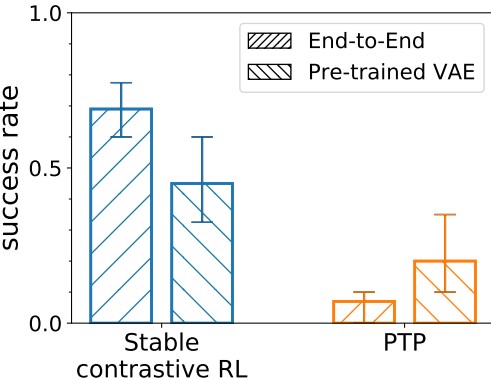 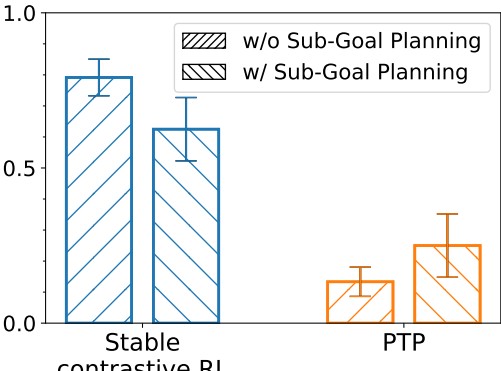

Figure 24: *(Left)* Additional perceptual losses are not needed for stable contrastive RL to outperform PTP. *(Right)* Without sub-goal planning, stable contrastive RL can solve temporal-extended tasks.

tion) and an end-to-end variant of PTP. We use the task `push block, open drawer` for these experiments, as it demands the most temporally-extended reasoning to solve.

We compare all methods by training three random seeds of each of the four methods for 300K gradient steps on the offline setting. We measure performance by evaluating each seed in 10 rollouts. Fig. 24 *(Left)* shows the mean and standard deviation of these results. First, we find that stable contrastive RL achieves $53\%$ higher success rates than the VAE-based version of contrastive RL, showing that contrastive RL does not require a perception specific loss, and instead suggesting that a good representation emerges purely from optimizing the contrastive RL objective. Intuitively, this result makes sense, as the contrastive RL critic objective already resembles existing representation learning objectives. The observation that adding the VAE objective *decreases* performance might be explained by the misalignment between the VAE objective and the contrastive RL objective – directly optimizing the representations for the target task (learning a critic) is better than optimizing them for reconstruction (as done by the VAE). We conjecture that VAE representation could contain task-irrelevant noise that impairs policy learning. Second, we find that the VAE objective is important for the prior method, PTP; removing the VAE objective and learning the representations end-to-end decreases performance by $55\%$. This result supports the findings of the original PTP paper (Fang et al., 2022a), while also illustrating that good representations can be learned in an end-to-end manner if the RL algorithm is chosen appropriately.

### F.12 SUB-GOAL PLANNING

Goal-conditioned RL often requires the agent to reason over long horizons. Prior works has proposed to combine critic learning with an explicit planner (Savinov et al., 2018; Eysenbach et al., 2019; Fang et al., 2022a), using the learned value network as distance function to generate sub-goals. These *semi-parametric* methods can work really well in practice, but it remains unclear why *fully-parametric* model-free GCRL algorithms failed to perform similarly given the same training data. We have shown that stable contrastive RL learns a representation preserving causal information (Fig. 6) and a policy rearranging representations in temporal order (Fig. 19). So *can fully-parametric methods achieve comparable or better performance to semi-parametric methods with proper design decisions?*

To answer this question, we compare stable contrastive RL (a fully-parametric algorithm) with a variant that use explicit sub-goal planning. We compare against PTP (a semi-parametric method) and a variant of PTP without sub-goal planning. To make a fair comparison, we apply the same planning algorithm of PTP to the variant of stable contrastive RL. We choose the task `pick & place (table)` which requires the agent to pick and place a block to conduct our experiments. We train three random seeds of all the four methods for 300K gradient steps with the same offline dataset and evaluate the success rate over 10 rollouts, reporting the mean and standard deviation across seeds.

As shown in Fig. 24 *(Right)*, stable contrastive RL achieves a $30\%$ absolute improvement over its semi-parametric variant, suggesting that stable contrastive RL does not require planner and, surprisingly, sub-goal planning could *hurt* the performance. We suspect that the policy can find a path in representation space that corresponds to an optimal trajectory in state space without additional

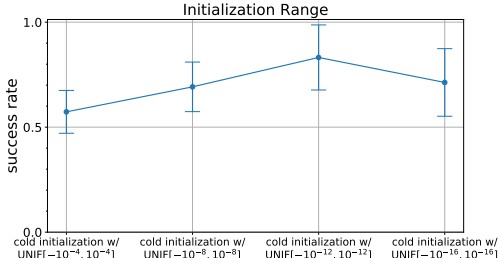

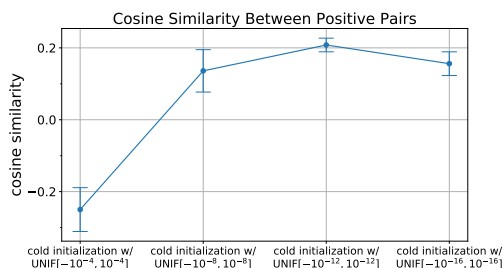

Figure 27: Smaller cold initialization (in a certain range) improves the performance of stable contrastive RL.

Figure 28: A fairly small cold initialization value encourages alignment.

supervision. The reason for adding planning module *decrease* performance might be explained by the fact that self-supervised critic objective does not imply a distance measure between the current state and the goal, and the growing computational complexity of planning. We observe that sub-goal planning is critical to improve the performance of PTP by $50\%$, which is consistent with the claim in the original PTP paper (Fang et al., 2022a). Nonetheless, neither version of PTP achieves comparable results to stable contrastive RL, showing that semi-parametric methods might not do well in a task with causal reasoning. Taken together, these experiments suggest that, with proper design decisions, a fully-parametric method like stable contrastive RL can at match (if not surpass) the performance of more complex semi-parametric methods that employ planning.

### F.13 ABLATION EXPERIMENTS: BATCH SIZE

These ablation experiments test whether the batch size can affect the performance of stable contrastive RL and whether an ever larger batch size can induce an ever improvement in performance. To answer these questions, we vary the batch size from $512$ to $4096$ during training of stable contrastive RL and find the performance follows the batch size increasing paradigm as expected. However, the improvement stops after a certain threshold, indicating there might be other factors, e.g. network capacity and training steps (Chen et al., 2020), that dominate the performance after a batch size limit.

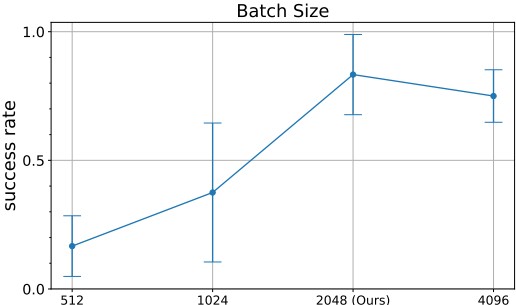

Figure 25: Larger batch size (in a certain range) improves the performance of stable contrastive RL.

### F.14 ABLATION EXPERIMENTS: COLD INITIALIZATION

We run additional experiments to study whether cold initialization provides similar performance gains as learning rate schemes. We compare three methods: *(a)* cold initialization as mentioned in Sec. 3.2. *(b)* learning rate warmup, following the same warmup paradigm in (Vaswani et al., 2017) – linearly increasing the learning rate to $3 \times 10^{-4}$ for the first 100K gradient steps and then decreasing it proportionally to the inverse square root of remaining gradient steps. *(c)* using a $10\times$ smaller learning rate for the last layers of $\phi(s, a)$ and $\psi(g)$. Our new experiments (Fig. 26) find that both *(b)* and *(c)* consistently perform worse than cold initialization.

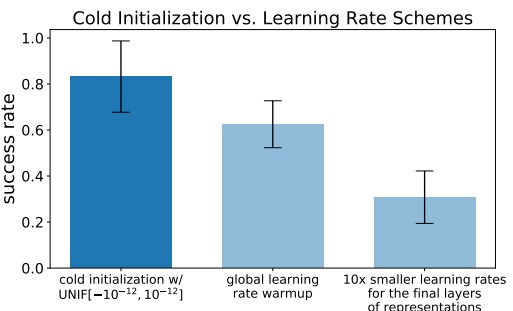

Figure 26: Cold initialization is more effective than learning rate schemes for stable contrastive RL.

To get a better understanding of the mechanism of cold initialization, we hypothesize that the range of initialization will affect the performance of stable contrastive RL. We conducted ablation experiments with initialization strategies $\text{UNIF}[-10^{-4}, 10^{-4}]$, $\text{UNIF}[-10^{-8}, 10^{-8}]$, $\text{UNIF}[-10^{-12}, 10^{-12}]$, and $\text{UNIF}[-10^{-16}, 10^{-16}]$ for the final layers of $\phi(s, a)$ and $\psi(g)$, plotting the success rates over 10 episodes and 3 random seeds in Fig. 27. The observation that small initialization ranges tend to perform better, with a value of $10^{-12}$ achieving $45\%$ higher success rate than a (more standard) range of $10^{-4}$, is consistent with our expectation that a fairly small initialization range incurs better performance. We note that the performance of the range $10^{-16}$ is $0.857\times$ ($71.3\%$ vs. $83.2\%$) than that of the range $10^{-12}$, suggesting that an ever decreasing initialization range start to hinder representation learning after a certain value.

One effect of the cold initialization is that it changes the average distance between representations at random initialization. We next measure the average pairwise (cosine) distance between representations, for different initialization ranges. We randomly sample 10 validation batches and compute the average cosine similarities of positive pair at initialization as a function of different cold initialization ranges, showing results in Fig. 28. The finding that cosine similarities between positive examples decrease when we vary the initialization ranges from $10^{-4}$ to $10^{-12}$ explains the intuition that alignment between positive examples is crucial to contrastive representation learning. Of particular note is that using a very small initialization range $10^{-16}$ could force all representations to collapse, resulting in a worse performance. We observe that initialization strategies for which the representations are closer (higher cosine similarity) also result in higher success rates after training. This suggests that the average pairwise distance at initialization may be an effective way of selecting the initialization range *that does not require performing any training*.

### F.15 ABLATION EXPERIMENTS: CONTRASTIVE REPRESENTATION DIMENSION

While we used small representations (16 dimensional) in the main experiments, we now study how increasing the representation dimension affects the success rates. We ablate the dimension of contrastive representation in the set $\{16, 128, 512\}$, averaging success rates over 10 episodes of 3 random seeds. As shown in Fig. 29, representations of sizes 128 and 512 achieve considerably lower success rates, echoing prior work in finding that smaller representations yield better performance in the offline setting (Eysenbach et al., 2022). These results might also be explained by the increase of noise with a larger representation size, suggesting that the smaller representations effectively act as a sort of regularization and mitigate overfitting.

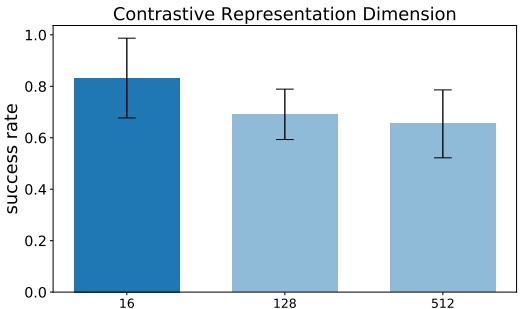

Figure 29: Smaller representation dimension benefits the learning of stable contrastive RL.

