# OpenReview forum: "Stabilizing Contrastive RL: Techniques for Robotic Goal Reaching from Offline Data"
_ICLR.cc/2024/Conference — ICLR 2024 spotlight_

### Official Review · Reviewer_8PDP · 2023-11-01

**Soundness:** 3 good
**Presentation:** 2 fair
**Contribution:** 2 fair
**Rating:** 5
**Confidence:** 4

**Summary:**

This paper delves into the stabilization of contrastive reinforcement learning (RL). Based on the paper "Contrastive learning as goal-conditioned reinforcement learning", the authors examine a variety of design elements, including:

- Neural network architecture for image inputs
- Batch size
- Layer normalization
- Cold initialization
- Data augmentation


Through empirical evaluations conducted on the `drawer` task, the authors give a set of conclusions regarding these design choices. They name this combination of design choices as "stable contrastive RL".

Furthermore, the paper compares the performance of their stable contrastive RL approach with previous methods, testing in both simulated and real-world scenarios. Additional experiments are also presented, showing distinct attributes of their proposed method.

The contributions of this paper go as follows:

- It conducts an in-depth investigation of various design choices based on the paper "Contrastive learning as goal-conditioned reinforcement learning". Through experiments on the drawer task, the authors provide empirical evidence supporting their conclusions on the aforementioned design choices.

- Beyond the primary focus, the authors also undertake additional experiments to further evaluate the properties and advantages of their proposed method.

**Strengths:**

- The paper conducts a detailed examination of various design choices in contrastive reinforcement learning, providing readers with a holistic understanding.

- By undertaking experiments on the drawer task, the paper grounds its conclusions in empirical evidence, adding credibility to its findings.

- By contrasting the proposed method with existing ones in both simulated and real-world environments, the paper offers a clear benchmark of its advantages and potential shortcomings.

**Weaknesses:**

While the paper presents a lot of experimental findings, it largely feels akin to an experimental report with uncorrelated results. The major motivation remains ambiguous, making it challenging to discern the story the authors aim to convey.

Specific weaknesses include:

- Motivation is not clear
	- The primary objective seems to be stabilizing contrastive RL. However, the question remains: Why is there a need to stabilize "contrastive RL" in the first place? The paper falls short in explaining the inherent instability of contrastive RL.
    - Even if the stabilization of contrastive RL is deemed necessary, the rationale for examining specific aspects like architecture and batch size remains unclear.

- Structure of the experiment section is messy
	- The experiment section, segmented into seven sub-sections, lacks clarity. The interrelation of these experiments is not explicitly explained, leaving readers questioning their significance.
	- Several experiments don't directly align with the paper's core focus on stabilizing contrastive RL. Queries arise such as:
		- Why introduce a comparison of pre-trained representation?
        - What is the relevance of latent interpolation?
		- How does the arm matching problem fit into the story?
		- What motivates the test on generalization across unseen camera poses and object color?
	- The presentation in section 4.2 further complicates comprehension. While five design decisions are discussed in section 3.2, their individual results aren not distinctly showcased in section 4.2, leading to confusion.

- Limited backbone algorithm
	- The entire experimental framework is based on the method "Contrastive learning as goal-conditioned reinforcement learning". This singular focus raises concerns about the generalizability of the conclusions.

- Limited Task Support for Major Conclusions
	- Major conclusions stem from the `drawer` task experiments (Sec 4.2). Even though additional tasks are addressed in the appendix, the overall spectrum of tasks remains restricted.
	- Why not conduct the abalation study on all the benchmarks mentioned in Sec 4.1?

**Questions:**

- See the weakness part of this review.
- In Sec 3.2, it mentions "using larger batches increases not only the number of positive examples (linear in batch size) but also the number of negative examples (quadratic in batch size)". Could you further explain this point?

---

> ### Author Response · Authors · 2023-11-17
> **Rebuttal by Authors - Part 1**
>
> We thank the reviewers for the detailed responses and for the suggestions for improving the paper. It seems like the main question is about the motivation and structure of experiments. We agree that this paper is an empirical report – it documents a year's worth of experiments aimed at building a better offline algorithm for goal-conditioned RL problems. We believe these findings may be of interest to some of the ICLR community that works on robotics, scaling offline RL, and related topics. We agree that the empirical flavor of these results may not be interesting to all facets of the ICLR community. **Together with the discussion below and the revision in the paper (orange texts), does this fully address the reviewer’s questions?**
>
> > Motivation is not clear
>
> Our core focus is on (a) building a better goal-conditioned RL algorithm that is applicable to robotic tasks, (b) making initial attempts to understand the difference between our method and prior approaches, (c) and empirically analyzing why our method works better than baselines. We have revised the introduction to clarify these motivations.
>
> Prior methods sharing similar goals discussed various design decisions that would benefit the performance of an RL algorithm in both online [1] and offline setting [2][3]. “Stability” turned out to be an important step toward these goals: we applied layer normalization to balance feature magnitude of different tasks (Sec. 4.2), and use data augmentation to prevent overfitting of the GCBC regularization (Appendix Fig. 18). Appendix Fig. 13 shows that our method is more stable (less oscillatory) than the original contrastive RL algorithm.
>
> In addition to proposing design decisions that improve performance of an existing algorithm, we also included an visualization of representations learned by different methods (Sec. 4.4) and a quantitative analysis of learned Q values in the discussion of the arm matching problem (Sec. 4.5), providing empirical explanations for why stable contrastive RL learned to solve the tasks while baselines failed to do so. These studies potentially motivate future work on learning structural representations for robotic tasks.
>
> > The experiment section, segmented into seven sub-sections, lacks clarity.
>
> To clarify the experiments, we have added a list of research questions we are trying to answer using our experiments at the beginning of the experiment section and also have updated headers of subsections of our experiments to refer to those questions. We have also revised this section to explain the relationships between the experiments.
>
> > Why introduce a comparison of pre-trained representation?
>
> Prior work [3][4] has used pre-trained off-the shelf visual encoder for an algorithm. We introduce this comparison to study whether pre-training the representation with an auxiliary objective boosts performance (it doesn't). We have added a sentence in the “Network architecture” paragraph of Sec. 4.2 to clarify this.
>
> > What is the relevance of latent interpolation?
>
> The aim of the latent interpolation experiment (Fig. 6) is to provide some intuition into the learned representations. We believe these visualizations are relevant to the discussion because they demonstrate that our method captures causal relationships in its representation space while baselines failed; this offers one hypothesis for why stable contrastive RL outperforms prior methods in Sec. 4.3. We have added a sentence at the end of Sec. 4.3 to clarify.
>
> > How does the arm matching problem fit into the story?
>
> The arm matching problem is a common failure mode of prior methods; we wanted to analyze this case to understand if and how our method was avoiding this failure case, achieving a higher success rate. The discussions in Sec. 4.5 and Appendix F.6 provide a quantitative analysis of the (normalized) Q values learned by our method and baselines, showing that stable contrastive RL achieves lower error in credit assignments. We believe these experiments offer another hypothesis for why stable contrastive RL outperforms prior methods in Sec. 4.3. We also revised the final paragraph of Sec. 4.3 to clarify this.
>
> > What motivates the test on generalization across unseen camera poses and object color?
>
> We have revised the introduction to clarify that we focus not only on studying design decisions that produce a better goal-conditioned RL algorithm based on contrastive RL, but also on investigating the properties of the policy learned by our method. Prior methods have used contrastive learning as auxiliary loss besides RL objective to improve the robustness of RL algorithm with image inputs [4][5]. This motivates us to study the robustness of the learned policy by testing the generalization of stable contrastive RL on unseen camera poses and object colors.

---

> ### Author Response · Authors · 2023-11-17
> **Rebuttal by Authors - Part 2**
>
> > While five design decisions are discussed in section 3.2, their individual results are not distinctly showcased in section 4.2, leading to confusion.
>
> We grouped those 5 design decisions (Sec. 3.2) into two main classes due to limited pages in Sec. 4.2. We have added labels (D1, D2, etc.) in Sec. 3.2 and Sec. 4.2 for each of the design decisions to clarify.
>
> > The entire experimental framework is based on the method "Contrastive learning as goal-conditioned reinforcement learning". This singular focus raises concerns about the generalizability of the conclusions.
>
> One of the aims of this paper is to find design decisions that improve an existing goal-conditioned RL algorithm based on contrastive learning. While we did indeed use this method as a starting point, we evaluate our resulting method by comparing it to 8 prior methods (Sec. 4.3), including those based on different backbone algorithms. Note that we have also run experiments to study the effect of applying some of components (e.g. auxiliary perception loss and sub-goal planning) from prior methods to our approach (Appendix F.11 & Appendix F.12).
>
> > Limited Task Support for Major Conclusions
>
> While Fig. 1 in the main text does the ablation on the $\texttt{drawer}$ environment, Fig. 14 in the appendix repeats these ablation experiments on the $\texttt{push block, open drawer}$ environment, drawing similar conclusions. We have revised the first paragraph of Sec. 4.2 to clarify. Because of compute constraints, it's not feasible to run the ablation on every environment. Nonetheless, we'd be happy to run additional narrow experiments to answer any targeted questions the reviewer may have.
>
> > In Sec 3.2, it mentions "using larger batches increases not only the number of positive examples (linear in batch size) but also the number of negative examples (quadratic in batch size)". Could you further explain this point?
>
> This point is derived from the original implementation of contrastive RL [6] and we have added a citation in this sentence to clarify. As mentioned in Sec. 3.1, the critic function is parameterized by the inner product between the state-action representation $\phi(s, a)$ and future state representation $\psi(s_{t+})$: $f(s, a, s_{t+}) = \phi(s, a)^{\top} \psi(s_{t+})$. In practice, we first sampled a batch of state-action representations $\\{\phi(s^{(i)}, a^{(i)})\\}^N_{i = 1}$ and the corresponding future state representations $\\{\psi(s_{t+}^{(i)})\\}^N_{i = 1}$, and then compute their $\textit{outer products}$ to construct a critic matrix $F \in \mathbb{R}^{N \times N}$ with $F[i, j] = \phi(s^{(i)}, a^{(i)})^{\top} \psi(s_{t+}^{(j)})$. The diagonal entries in the critic matrix $F$ are positive pairs while the off-diagonal entries in the critic matrix $F$ are negative pairs. Thus, increasing the batch size $N$ results in a linear increase in positive examples and a quadratic increase in negative examples.
>
> [1] Fujimoto, S., Hoof, H. and Meger, D., 2018, July. Addressing function approximation error in actor-critic methods. In International conference on machine learning (pp. 1587-1596). PMLR.
>
> [2] Ball, P.J., Smith, L., Kostrikov, I. and Levine, S., 2023. Efficient online reinforcement learning with offline data. arXiv preprint arXiv:2302.02948.
>
> [3] Kumar, A., Agarwal, R., Geng, X., Tucker, G. and Levine, S., 2022, September. Offline Q-learning on Diverse Multi-Task Data Both Scales And Generalizes. In The Eleventh International Conference on Learning Representations.
>
> [4] Nair, S., Rajeswaran, A., Kumar, V., Finn, C. and Gupta, A., 2023, March. R3M: A Universal Visual Representation for Robot Manipulation. In Conference on Robot Learning (pp. 892-909). PMLR.
>
> [5] Laskin, M., Lee, K., Stooke, A., Pinto, L., Abbeel, P. and Srinivas, A., 2020. Reinforcement learning with augmented data. Advances in neural information processing systems, 33, pp.19884-19895.
>
> [6] Eysenbach, B., Zhang, T., Levine, S. and Salakhutdinov, R.R., 2022. Contrastive learning as goal-conditioned reinforcement learning. Advances in Neural Information Processing Systems, 35, pp.35603-35620.

---

> > ### Author Response · Authors · 2023-11-21
> > **Do the revisions address the concerns?**
> >
> > Dear Reviewer,
> >
> > **Do the revisions described above fully address the concerns about the paper?** We believe that these changes further strengthen the paper and help to clarify its claims. While the time remaining in the review period is limited, we would be happy to try to run additional experiments or make additional revisions.
> >
> > Kind regards,
> >
> > The Authors

---

> > > ### Comment · Reviewer_8PDP · 2023-11-22
> > > **Thank the authors for the detailed explanation**
> > >
> > > Thank the authors for the detailed explanation. Now I feel the structure of the experiments section is more clear, and I have understood the objectives of most experiments.
> > >
> > >
> > >
> > > My remaining concerns:
> > >
> > > 1. The authors have not directly answered my question about the motivation: Why is there a need to stabilize "contrastive RL" in the first place? The paper falls short in explaining the inherent instability of contrastive RL.
> > >
> > > 2. I entirely understand it would be expensive to run all ablation studies and experiments on all tasks. However, the authors did not provide sufficient evidence about why `drawer` is a representative task to run ablations.
> > >
> > >
> > >
> > > I am willing to raise my rating if my concerns can be resolved.

---

> ### Author Response · Authors · 2023-11-22
> **Author response**
>
> Author response
>
> Thanks for clarifying that the rebuttal addresses the concerns about organization and objectives of the experiments, but not the concerns about why contrastive RL needs stabilization and why drawer is a representative task to use in the ablations.
>
> We have further revised the paper to address these two points. We describe the changes below:
>
> > evidence about why `drawer` is a representative task to run ablations.
>
> The ablation experiments are done on two tasks: $\texttt{drawer}$ and $\texttt{push block, open drawer}$, both using image-based observations. Here's why we selected these two tasks:
> * While using state-based tasks would have been computationally less expensive, we opted for the image-based versions of the tasks because they more closely mirror the real-world use case (Fig. 5), where the robot has image-based observations.
> * Prior methods struggle to solve both these tasks, including (original) contrastive RL (Fig. 2 (Right)). Thus, there was ample room for improvement.
> * Out of the tasks in Figure 2, we choose the easiest task ($\texttt{drawer}$, which is still challenging for baselines) and one of the most complex tasks ($\texttt{push block, drawer open}$).
>
> Of course, the most representative task would have been to ablate these decisions on the physical robot, but that would have been extraordinarily expensive. We believe that the two representative tasks used are a good faith effort to evaluate our design decisions. We have added this discussion to Appendix F.4.
>
> If it would help to address the concern, we would be happy to run an additional ablation experiment on another one of the tasks.
>
> > Why is there a need to stabilize "contrastive RL" in the first place?
>
> Perhaps “contrastive RL v2” or “improved contrastive RL” would be a more clear name for the method. We called our method “stable contrastive RL” because, when we looked at the learning curves (Fig. 13), they were smoother and had much lower variance across random seeds. We would be happy to update the method / title if the reviewer thinks that would address this potential for confusion.

---

> > ### Comment · Reviewer_8PDP · 2023-11-23
> >
> > Thank you for the clarification. I have raised my score.

---

### Official Review · Reviewer_wQCi · 2023-11-01

**Soundness:** 4 excellent
**Presentation:** 4 excellent
**Contribution:** 3 good
**Rating:** 8
**Confidence:** 3

**Summary:**

This paper proposes a method, stable contrastive RL, that is a variant of contrastive RL with boosted performances over previous contrastive RL and goal-conditioned RL methods.
They aim to tackle with real-world robotic tasks where previous contrastive RL methods have not been applied
But applying contrastive RL, self-supervised version of goal-conditioned RL, to real-world robotic tasks faces several challenges.
Self-supervised learning have been  making innovations in CV and NLP, but those recent innovations may not transfer to the RL setting
Starting from design decisions  regarding model capacity and regularization, they found important and concrete design factors, such as the architecture, batch size, normalization, initialization, and augmentation, through intensive experiments.
They achieved +45% performance over prior implementations of contrastive RL, and 2× performance relative to alternative goal- conditioned RL methods in simulated environments. Finally, they showed  that these design decisions enable image-based robotic manipulation tasks in real-world experiments.

**Strengths:**

The authors use clear descriptions throughout the paper. Backgrounds,  related works, the definition of  the problem are all clear and sufficiently described and detailed.
In previous works heuristically designed visual representations are used without detailed designs and experiments of the visual  representations. The authors did  a great contribution to this field that they did a detailed designs of architectures and algorithms with intensive experiments. Those experiments support their claims well.

**Weaknesses:**

The authors did great experiments with variety of tasks with intensive analysis from multiple viewpoints.
More analytical descriptions in performance comparisons would contribute more to RL fields.
For example, in the simulation analysis of manipulation section, they used the simple sentence analysis," perhaps because the block in that task occludes the drawer handle and introduces partial observability. ", for worse performance.
Also, while pages are limited, some analysis of learned representations of stable contrastive RL in relation to performance combined with visual situations would help to understand those performance comparison analysis .

**Questions:**

After reading through the paper, I understand the one of the aims and contributions of the paper is to apply contrastive RL to real-world robot applications. But most of the paper are descriptions for simulated situations. This may blur your aims and contributions.

---

> ### Author Response · Authors · 2023-11-17
> **Rebuttal by Authors**
>
> We thank the reviewer for the responses and suggestions for improving the paper. We have revised the paper based on these suggestions (orange texts) and describe in more detail below.
>
> > … some analysis of learned representations of stable contrastive RL in relation to performance combined with visual situations would help to understand those performance comparison analysis.
>
> We agree that analysis of learned representations of stable contrastive RL in addition to simply comparing performance would help understand the difference between our method and baselines. As an initial step in this direction, we have visualized the representations learned by our approach and baselines in Sec. 4.4 and Appendix F.5, showing that stable contrastive RL captures causal relationships in its representation space while baselines (GCBC) failed to do so. Quantitatively, we measure the alignment of the interpolated representations with respect to the ground truth sequences in Appendix F.6, showing that stable contrastive RL achieves lower error than the alternative methods. Additionally, Appendix F.7 contains a comparison of the normalized Q learned by stable contrastive RL to those learned by baselines through an optimal trajectory, suggesting that stable contrastive RL assigns values to different observations correctly over baselines. These experiments provide empirical explanations for why stable contrastive RL outperforms prior methods. We have revised Sec. 4.3 to add references to these empirical analysis. We welcome additional suggestions for analyzing the performances.
>
> > But most of the paper are descriptions for simulated situations. This may blur your aims and contributions.
>
> Thanks for the suggestion. We have added this into the limitation section.

---

### Official Review · Reviewer_kFCy · 2023-11-02

**Soundness:** 3 good
**Presentation:** 3 good
**Contribution:** 3 good
**Rating:** 8
**Confidence:** 4

**Summary:**

Contrastive reinforcement learning is a method for learning goal-conditioned policies from offline data. Prior works have had moderate success with training RL agents using contrastive learning; this paper increases the success rate of contrastive RL agents and deploys them in more domains. To that end, the paper describes a set of techniques, like choosing the right network size and changing the initialization of the last network layer, that greatly improve the agent's success rate. The method is tested in simulated and real-world robotic manipulation tasks as well as in a simulated locomotion task.

**Strengths:**

1. The paper clearly describes several tricks that greatly increase the success rate of contrastive RL agents. In particular, it is interesting that the final layer initialization trick does better than learning rate warm-up.
2. The experiments include a large number of relevant baselines, including baselines that are pre-trained on large video datasets.
3. The authors identify the “arm matching problem”, where the value function cares about the state of the robot arm but not the environment.

**Weaknesses:**

1. The paper demonstrates that contrastive RL is a brittle objective, since small changes in the network architecture and initialization lead to huge changes in agent performance. The paper does not study if the objective could be changed to make it more robust and less prone to overfitting.

2. It is unclear why none of the methods can learn, e.g., the “push can” policy in Figure 5. Overall, the paper does not do a good job of explaining why some of the seemingly simple tasks are difficult and why SOTA robot learning methods cannot solve them.

Minor:
* The second and third paragraph of the introduction could be more specific. E.g. “various design decisions” is too ambiguous.
* “However, to the best of our knowledge these contrastive RL methods have not been applied on real-world robotic systems.” – https://arxiv.org/abs/2306.00958, https://arxiv.org/abs/2203.12601 use contrastive learning with offline RL and are deployed on real-world systems. This claim is only true if we use a very narrow definition of “contrastive RL methods”.

**Questions:**

1. Could standard imitation learning approaches that use point clouds (like PerAct https://arxiv.org/abs/2209.05451, RVT https://arxiv.org/abs/2306.14896 and Act3D https://arxiv.org/abs/2306.17817) be able to solve the tasks in Figure 5 with high success rate?

2. Why is it difficult to reach high success rates on the seemingly simple robot tasks in Figure 2 and 5?

---

> ### Author Response · Authors · 2023-11-17
> **Rebuttal by Authors**
>
> We thank the reviewer for the detailed responses and helpful suggestions for improving the paper. The reviewer raised a number of good questions about the paper, which we have attempted to address below and which we have already incorporated into the paper (orange texts). We believe that these revisions to the paper make it stronger. **Together with the discussions below, does this fully address the reviewer’s concerns?**
>
> > Could standard imitation learning approaches that use point clouds (like PerAct https://arxiv.org/abs/2209.05451, RVT https://arxiv.org/abs/2306.14896 and Act3D https://arxiv.org/abs/2306.17817) be able to solve the tasks in Figure 5 with high success rate?
>
> Comparing to imitation learning methods using point clouds as inputs may take some time, as the standard benchmarks we've used don't support point cloud observations. One way of interpreting the results in Fig 3 is that the gap between representation-based and image-based results suggests that there is room for more sophisticated representation learning methods (e.g., based on point clouds) to boost performance. We have added a discussion of point-cloud based methods (including [1][2][3]) to the “Representation Learning in RL” paragraph of the related work section.
>
> > Overall, the paper does not do a good job of explaining why some of the seemingly simple tasks are difficult … Why is it difficult to reach high success rates on the seemingly simple robot tasks in Figure 2 and 5?
>
> We have discussed the challenges in solving these tasks in Sec. 4.1 and Appendix E. Our experiments use a suite of simulated goal-conditioned control tasks based on prior work [4] (Fig. 2 left). These tasks are challenging because we directly use the 48 x 48 x 3 RGB images as observations and goals and solving most of them requires multi-stage reasoning. For example, the optimal trajectory for $\texttt{push block, open drawer}$ includes first pushing the orange block away and then opening the drawer. Similarly, to solve the task $\texttt{pick \\& place (drawer)}$, the policy needs to pick and place the green block first and then close the drawer. Note that the orders of completing those stages (sub-tasks) cannot change, suggesting that an algorithm achieving high success rates should be able to figure out the causal relationship. For the real-world manipulation and locomotion benchmarks, these tasks are difficult because (a) the offline datasets used for training contain suboptimal trajectories missing the objects or target poses[5], (b) evaluation goals are not revealed during training, (c) and the policy learns to solve different evaluation tasks by learning on one large offline dataset.
>
> > … why SOTA robot learning methods cannot solve them.
>
> As an initial step in explaining the failure of baselines, we have visualized the representations learned by our approach and other baselines in Sec. 4.4 and Appendix F.5, showing that stable contrastive RL captures causal relationships in its representation space while baselines (GCBC) failed to do so. Quantitatively, we measure the alignment of the interpolated representations with respect to the ground truth sequences in Appendix F.6, showing that stable contrastive RL achieves lower error than the alternative methods. Additionally, Appendix F.7 contains a comparison of the normalized Q learned by stable contrastive RL to those learned by baselines through an optimal trajectory, suggesting that stable contrastive RL assigns values to different observations correctly over baselines. These experiments provide empirical explanations for why stable contrastive RL outperforms prior methods. We have revised Sec. 4.3 to add references to these empirical analysis.
>
> > The paper does not study if the objective could be changed to make it more robust and less prone to overfitting.
>
> We agree that our focus is on one contrastive RL objective. However, our experiments do compare to prior methods that use different goal-conditioned RL objectives, e.g., advantage weighted policy regression (WGCSL [6]). In particular, VIP [7] and R3M [8] also use the idea of contrastive learning to obtain representations from large real-world robotic datasets, potentially more robust and less prone to overfitting. We compared to GCBC variants trained on top of those learned representations in Sec. 4.3 and found that stable contrastive RL still outperformed them on 4 / 5 tasks.
>
> > The second and third paragraph of the introduction could be more specific. E.g. “various design decisions” is too ambiguous.
>
> We have revised this to clarify.
>
> > “However, to the best of our knowledge these contrastive RL methods have not been applied on real-world robotic systems.” … This claim is only true if we use a very narrow definition of “contrastive RL methods”.
>
> We have removed this sentence in the introduction.

---

> > ### Author Response · Authors · 2023-11-17
> > **References**
> >
> > [1] Shridhar, M., Manuelli, L. and Fox, D., 2023, March. Perceiver-actor: A multi-task transformer for robotic manipulation. In Conference on Robot Learning (pp. 785-799). PMLR.
> >
> > [2] Goyal, A., Xu, J., Guo, Y., Blukis, V., Chao, Y.W. and Fox, D., 2023. RVT: Robotic View Transformer for 3D Object Manipulation. arXiv preprint arXiv:2306.14896.
> >
> > [3] Gervet, T., Xian, Z., Gkanatsios, N. and Fragkiadaki, K., 2023, August. Act3D: 3D Feature Field Transformers for Multi-Task Robotic Manipulation. In 7th Annual Conference on Robot Learning.
> >
> > [4] Fang, K., Yin, P., Nair, A. and Levine, S., 2022, October. Planning to practice: Efficient online fine-tuning by composing goals in latent space. In 2022 IEEE/RSJ International Conference on Intelligent Robots and Systems (IROS) (pp. 4076-4083). IEEE.
> >
> > [5] Walke, H.R., Black, K., Zhao, T.Z., Vuong, Q., Zheng, C., Hansen-Estruch, P., He, A.W., Myers, V., Kim, M.J., Du, M. and Lee, A., 2023, August. BridgeData V2: A Dataset for Robot Learning at Scale. In 7th Annual Conference on Robot Learning.
> >
> > [6] Yang, R., Lu, Y., Li, W., Sun, H., Fang, M., Du, Y., Li, X., Han, L. and Zhang, C., 2021, October. Rethinking Goal-Conditioned Supervised Learning and Its Connection to Offline RL. In International Conference on Learning Representations.
> >
> > [7] Ma, Y.J., Sodhani, S., Jayaraman, D., Bastani, O., Kumar, V. and Zhang, A., 2022, September. VIP: Towards Universal Visual Reward and Representation via Value-Implicit Pre-Training. In The Eleventh International Conference on Learning Representations.
> >
> > [8] Nair, S., Rajeswaran, A., Kumar, V., Finn, C. and Gupta, A., 2022, August. R3M: A Universal Visual Representation for Robot Manipulation. In 6th Annual Conference on Robot Learning.

---

> > ### Author Response · Authors · 2023-11-21
> > **Do the revisions address the concerns?**
> >
> > Dear Reviewer,
> >
> > **Do the revisions described above fully address the concerns about the paper?** We believe that these changes further strengthen the paper. While the time remaining in the review period is limited, we would be happy to try to run additional experiments or make additional revisions.
> >
> > Kind regards,
> >
> > The Authors

---

> > > ### Comment · Reviewer_kFCy · 2023-11-22
> > > **Response**
> > >
> > > Thank you for answering my questions. I have raised my score.

---

### Official Review · Reviewer_8bYx · 2023-11-06

**Soundness:** 3 good
**Presentation:** 4 excellent
**Contribution:** 3 good
**Rating:** 8
**Confidence:** 4

**Summary:**

This paper aims to deploy contrastive reinforcement learning on real-world offline data. Towards this goal, the authors study various different design decisions in contrastive RL to improve its stability and performance. They consider architecture, data augmentation, and initialization.

They examine performance on simulated manipulation tasks and find a combination that outperforms prior work. They also compare against methods that use dedicated representation learning auxiliary losses and find that their stable contrastive RL outperforms these methods.

Next, they use stable contrastive RL to learn goal-directed image-based manipulation tasks solely from real-world offline data. They find that it is more effective than baselines.

Next, they examine the representation learned by contrastive RL compared to self-supervised methods and find that the representations learned by contrastive RL better capture the world dynamics.

Additional experiments that examine success misclassification, scaling laws, and generalization are also included.

**Strengths:**

The proposed method is effective and cleanly applies various improves from the literature to contrastive RL to improve its performance.

The goal of improve real-world performance is shown by real-world experiments.

There are multiple additional experiments that provided additional insight and the reviewer found them to be useful additions.

The appendix contains many useful details and experiments.

**Weaknesses:**

The conclusion that a deeper CNN performs worse than a shallow one, likely because of overfitting, indicates that maybe the benchmark tasks used do not align with the paper's goal of leveraging a vast amount of unlabeled data, like current approaches in computer vision and NLP.

There are some missing citations of RL references for the design decisions. For instance, McCandlish et al and Bjorck et al showed that large batch training and layer normalization, respectively, are effective in RL.

A full grid-search of all the combinations of design decisions would be great to have, but the reviewer understands that this may not be feasible.

### References

McCandlish et al, An Empirical Model of Large-Batch Training, 2018
Bjorck et al, Towards Deeper Deep Reinforcement Learning with Spectral Normalization, NeurIPS 2021

**Questions:**

None

---

> ### Author Response · Authors · 2023-11-17
> **Rebuttal by Authors**
>
> We thank the reviewers for the responses and suggestions for improving the paper. We have revised the paper based on these questions (orange texts), which we describe in more detail below.
>
> > The conclusion that a deeper CNN performs worse than a shallow one, likely because of overfitting, indicates that maybe the benchmark tasks used do not align with the paper's goal of leveraging a vast amount of unlabeled data, like current approaches in computer vision and NLP.
>
> We have revised the introduction to emphasize that our aim is to get goal-conditioned RL methods working on the real-world robotics datasets that we have today. While these datasets are big from a robotics perspective, they are admittedly much smaller than the datasets in CV / NLP (potentially explaining why we still see overfitting).
>
> > There are some missing citations … McCandlish et al and Bjorck et al …
>
> We have added these citation to the “Layer normalization” paragraph of Sec. 3.2.
>
> > A full grid-search of all the combinations of design decisions would be great to have, but the reviewer understands that this may not be feasible.
>
> We agree this is computationally infeasible, but would be happy to brainstorm and run more targeted experiments to answer directed questions.

---

### Meta-Review · Area_Chair_9EcE · 2023-12-06

**Metareview:**

The paper provides a concrete set of design decisions that make a variant of contrastive reinforcement learning useful in diverse and challenging environments. The empirical results show a substantial improvement over existing methods. Ablations are used to show the importance of various aspects of the proposed approach.

**Justification For Why Not Higher Score:**

The proposed technique is presented with some 'tricks' to make it work, however there does not seem to be a theoretical justification for why these tricks work. The tricks are only justified with empirical ablations.

**Justification For Why Not Lower Score:**

The paper presents a mechanism and design choices that make contrastive reinforcement learning work in many challenging environments which is noteworthy, especially considering the substantial performance improvements compared to prior work.

---

### Decision · Program_Chairs · 2024-01-16

Accept (spotlight)